# Atmospheric biogenic volatile organic compounds in the Alaskan Arctic tundra: constraints from measurements at Toolik Field Station

Vanessa Selimovic[1], Damien Ketcherside[1], Sreelekha Chaliyakunnel[1], Catie Wielgasz[1], Wade Permar[1],

Hélène Angot[2,*], Dylan B. Millet[3], Alan Fried[2], Detlev Helmig[4], Lu Hu[1]

[1]University of Montana, Missoula, MT, US
[2]University of Colorado Boulder, Institute of Arctic and Alpine Research, Boulder, CO, United States
[3]University of Minnesota Twin Cities, St Paul, MN, US
[4]Boulder A.I.R LLC, Boulder, CO, United States

* now at Ecole Polytechnique Fédérale de Lausanne (EPFL), Extreme Environments Research Laboratory, Sion, Switzerland

*Correspondence:* Lu Hu (lu.hu@mso.umt.edu)

**Abstract.** The Arctic is a climatically sensitive region that has experienced warming at almost three times the global average rate in recent decades, leading to an increase in Arctic greenness and a greater abundance of plants that emit biogenic volatile organic compounds (BVOCs). These changes in atmospheric emissions are expected to significantly modify the overall oxidative chemistry of the region and lead to changes in VOC composition and abundance, with implications for atmospheric processes. Nonetheless, observations needed to constrain our current understanding of these issues in this critical environment are sparse. This work presents novel atmospheric *in-situ* proton transfer reaction time-of-flight mass spectrometry (PTR-ToF-MS) measurements of VOCs at Toolik Field Station (TFS, 68º38' N, 149º36'W), in the Alaskan Arctic tundra during May-June 2019. We employ a custom nested grid version of the GEOS-Chem chemical transport model, driven with MEGANv2.1 (Model of Emissions of Gases and Aerosols from Nature version 2.1) biogenic emissions for Alaska at 0.25 ° × 0.3125° resolution, to interpret the observations in terms of their constraints on BVOC emissions, total reactive organic carbon (ROC) composition, and calculated OH reactivity (OHr) in this environment. We find total ambient mole fraction of 78 identified VOCs to be $6.3 \pm 0.4$ ppbv ($10.8 \pm 0.5$ ppbC), with overwhelming (>80%) contributions are from short-chain oxygenated VOCs (OVOCs) including methanol, acetone, and formaldehyde. Isoprene was the most abundant terpene identified. GEOS-Chem captures the observed isoprene (and its oxidation products), acetone, and acetaldehyde abundances within the combined model and observation uncertainties (±25%), but underestimates other oxygenated VOCs including methanol, formaldehyde, formic acid, and acetic acid by a factor of 3 to 12. The negative model bias for methanol is attributed to underestimated biogenic methanol emissions for the Alaska tundra in MEGANv2.1. Observed formaldehyde mole fractions increase exponentially with air temperature, likely reflecting its biogenic precursors and pointing to a systematic model underprediction of its secondary production. The median campaign calculated OHr from VOCs measured at Toolik was $0.7$ s$^{-1}$, roughly 5% of the values typically reported in lower-latitude forested ecosystems. Ten species account for over 80% of the calculated VOC OHr, with formaldehyde, isoprene, and acetaldehyde together accounting for nearly half of the total. Simulated OHr based on median

modelled VOCs included in GEOS-Chem averages 0.5 s$^{-1}$ and is dominated by isoprene (30%) and monoterpenes (17%). The data presented here serve as a critical evaluation of our knowledge of BVOCs and ROC budgets in high latitude environments and represent a foundation for investigating and interpreting future warming-driven changes in VOC emissions in the Alaskan Arctic tundra.

**1 Introduction**

The Arctic is a climatically sensitive region that has experienced temperature increases at almost three times the global average rate in the past century (AMAP, 2021; Post et al., 2019; Hansen et al., 2010). This rapid warming has increased Arctic greenness to include a larger abundance of shrubs and graminoids in the tundra ecosystem over the last few decades (Frost et al., 2020; Lindwall et al., 2016; Rinnan et al, 2014; Koesselmeier and Staudt, 1999). Similarly, present woody cover in the

Arctic is predicted to increase over 50% by 2050, which will amplify warming due to decreased surface albedo (Pearson et al., 2013; Guenther et al., 2012; Rinnan et al., 2011). These ecological changes are expected to increase emissions of biogenic volatile organic compounds (BVOCs) including isoprene and monoterpenes, which are emitted by plants partially in response to abiotic factors such as temperature and sunlight. Many other BVOCs are oxygenated (OVOCs), including alcohols, aldehydes, ketones, and organic acids. OVOCs are ubiquitous in the atmosphere and often have both direct biogenic sources

and photochemical sources, but their global budgets are poorly constrained, in part due to sparse availability of observational data. Among other factors, continued increases in warming have the potential to create positive feedback cycles associated with BVOC emissions, with likely impacts on tropospheric oxidative capacity in the Arctic related to ozone production and formation of secondary species. Though boreal, temperate, and tropical vegetation ecosystems have been surveyed for emission potentials of various BVOCs, observations are lacking to constrain BVOC emissions and their chemical impact in the highly

sensitive and changing Arctic tundra ecosystem. Quantifying changes in Arctic VOC emissions and evaluating model predictions thus requires high-quality baseline data along with an accurate understanding of the underlying processes driving VOC emissions in the region.

Global emission inventories assume BVOC fluxes in the Arctic to be minimal (~5% of total global isoprene and monoterpene

fluxes, despite being 18% of total global land area) due to lower average temperatures, shorter growing seasons, sparse vegetation cover, and lower basal emission factors in Arctic plants than compared with those in low and midlatitudes (Kramshøj et al., 2016; Sindelarova et al., 2014; Guenther et al., 2012). Field experiments focused on the warming effects on BVOC emissions have often observed stronger temperature sensitivity of Arctic and subarctic vegetation emissions than those in the lower latitudes (Angot et al., 2020; Lindwall et al., 2016; Kramshøj et al., 2016; Potosnak et al., 2013; Faubert et al.,

2010). These field observations often suggest a higher emission response to increased ambient temperature than predicted by BVOC emission inventories, which are generally based on responses to light and temperature among other environmental variables (Tang et al., 2016; Kramshøj et al., 2016; Potosnak et a., 2013; Guenther et al., 2012; Faubert et al., 2010). Studies have found that a steeper model temperature dependence yields isoprene emission rates more consistent with observations

(Tang et al., 2016). More recently, Angot et al., (2020) found a 180%-215% increase in isoprene emissions from Alaskan tundra vegetation in response to a 3-4°C warming, similar to increases predicted by a commonly used biogenic model for the 0-30°C temperature range (Guenther et al., (2012) (Model of Emissions of Gases and Aerosols from Nature version 2.1, or MEGANv2.1). These studies highlight the extreme temperature sensitivity of BVOC emissions from Arctic tundra ecosystems.

The limited number of previous model evaluation studies of high latitude atmospheric chemistry have mostly utilized short periods or 'snapshots' by aircraft field observations, but they have helped to identify knowledge gaps in our current understanding of OVOC budgets in the Arctic. For instance, a recent study coupling the GEOS-Chem chemical transport model (CTM) to observations from the Atmospheric Tomography (ATom) aircraft mission found underestimations in remote methanol abundance by over 50% in simulations from the base model. This underestimation was largest in the Arctic (>70%), except during wintertime, likely reflecting model errors in biogenic sources (Bates et al., 2021). Early intercomparisons of model results to surface observations have shown that CTMs have notable limitations in accurately simulating Arctic tropospheric composition, and that some of the largest discrepancies among models are found for OVOCs such as acetaldehyde and acetone. In one case, spring and summertime concentrations of acetaldehyde and acetone were both underestimated by CTMs (10-100% negative bias depending on the model) (Emmons et al., 2015). Other research has shown that biogenic emission inventories such as MEGAN overestimate acetone and its precursors in high latitudes (Wang et al., 2020). However, biogenic emissions are thought to only play a minor role (<10%) in formaldehyde vertical column densities observed from various observational platforms in Alaska during boreal summer, while methane oxidation (>60%) and wildfires (15%) are implied as more important sources (Zhao et al., 2022).

Emissions of formic and acetic acid are critical contributors to cloud water acidity in remote regions (Paulot et al., 2011). However, despite in-situ measurements at high latitudes showing mixing ratios of over 1 ppb for formic and acetic acid, modeled concentrations for both acids in the Arctic are very low (several ppt or less) (Mungall et al., 2018). Several explanations for this discrepancy have been suggested, including a direct biogenic source, and photochemical production from anthropogenic, biogenic and fire sources (Chen et al., 2021; Alwe et al., 2019; Schobesberger et al., 2016; Millet et al., 2015; Stavrakou et al., 2012). Recently, chamber studies by Franco et al., 2021 report efficient production of formic acid from formaldehyde via a multiphase reaction pathway that involves the hydrated form of formaldehyde, methanediol, in warm cloud droplets. The results mentioned above highlight the limited observational constraints and potential knowledge gaps of OVOC sources in high latitudes.

We note that some of these species are photochemically-interrelated and therefore enhancements and underestimation in one species are likely correlated with those from another. For example, reactions of isoprene and its oxidation products methacrolein (MACR) and methyl vinyl ketone (MVK) will readily produce formaldehyde via reactions with OH, as will oxidation of methanol and acetaldehyde. Reactive organic carbon (ROC) is expected to consist of hundreds of compounds

which can contribute to the formation of secondary species (Heald and Kroll., 2020). However, only a subset of these compounds is routinely measured, and an even smaller subset is modelled. As a result, our understanding of ROC abundance, distribution, and chemical impact remains poor for Arctic environments. In addition to the commonly studied VOCs mentioned earlier, recent studies utilizing advanced mass spectrometry instrumentation suggest that there are at least hundreds of organic compounds undergoing exchange between ecosystems and atmosphere (Goldstein and Galbally, 2007). Current CTMs do not account for that many species and are thought to underestimate ROC and reactivity as a result. Comparison to flux measurements in a mixed temperate forest indeed reveals that GEOS-Chem under-predicts total VOC carbon and reactivity by 40-60% on average, and these fluxes are dominated by compounds already explicitly included in the CTM. The results of this study suggest that the largest unknowns surrounding simulations of VOC carbon and reactivity in mixed temperate forests are associated with known, rather than unaccounted species (Millet et al., 2018), but to date no one has probed this critical issue in Arctic tundra environments.

This work presented here builds upon Angot et al. (2020) and showcases novel *in situ* proton-transfer-reaction time-of-flight mass-spectrometer (PTR-ToF-MS) ambient measurements of the entire VOC mass spectrum and a suite of other chemical and meteorological parameters at Toolik Field Station (TFS) in the Alaskan North Slope in the early summer of 2019. We compare observed mixing ratios of several major VOCs, and their temperature dependencies, with GEOS-Chem+MEGANv2.1 predictions, to identify if there are any key knowledge gaps for reactive carbon in the Arctic. Additionally, we investigate the full mass spectrum and identify contributions from previously unaccounted VOCs, as well as their potential to impact regional oxidative chemistry and estimates of total VOC carbon and OH reactivity.

## 2 Methods

### 2.1 Study site

Ambient VOC, nitrogen oxides ($NO_x$, where $NO_x = NO + NO_2$), $O_3$, and meteorological measurements were conducted from a weatherproof shelter roughly 350 m to the west of the base camp of Toolik Field Station (TFS) from 23 May to 23 June 2019. TFS is a long-term ecological research center located in the Arctic tundra on the northern flank of the Brooks Range in Northern Alaska (68º38' N, 149º36' W), roughly 178 km southwest of Prudhoe Bay (pop. ~2 k), and 600 km north of Fairbanks. The site is located ~250 km north of the Arctic Circle and is at an average elevation of 720 m above sea level. The Trans-Alaska Pipeline system and the Dalton highway, which run from north to south, are approximately 2 km to the east of the site. This area is typical of the northern foothills of the Brooks Range, with vegetation at this site largely categorized as tussock tundra within ~75 km radius (Angot et al., 2020, Elmendorf et al., 2012; Kade et al., 2012; Shaver and Chapin, 1991; Survey, 2012; Walker et al., 1994). Common plant species at the site include deciduous shrubs such as *Betula* (birch) and *Salix* (willow), as well as grasses such as *Eriphorium vaginatum* (cotton grass), and moss such as *Sphagnum angustifolium* (peat moss)(Angot et al., 2020).

## 2.2 Meteorological data

Figure 1 shows meteorological conditions at TFS during the monitoring period, measured from a meteorological tower located ~ 30 m from the instrument shelter (Angot et al., 2020). Average wind speed was 2.8 m/s, with a maximum of 9.0 m/s. Wind was primarily from the north and south, with occasional influences from the northwest (lake) and northeast (camp). Average hourly temperature for the entire study was roughly 7.5ºC and ranged from a minimum of -2.8ºC to a maximum of approximately 21ºC. A 10-year average of temperatures for this area suggests typical daily ranges of -6 ºC to 10 ºC between May and June. This range, and our campaign average reflects the seasonal transition, as the field intensive started near the onset of snowmelt (mid-May) and extended into the early growing season (mid-June). Both surface air temperature and photosynthetically active radiation (PAR) had distinct diurnal cycles, peaking between roughly 10:00 and 15:00 local standard time (AKST).

## 2.3 Proton transfer reaction time-of-flight mass spectrometer (PTR-ToF-MS)

Ambient VOC mixing ratios were measured by proton transfer time-of-flight mass spectrometry (PTR-ToF-MS 4000, Ionicon Analytik GmbH, Innsbruck, Austria). Air was pulled continuously from a sample inlet located 4 m above ground on a meteorological tower to the instrument at 10-15 L min$^{-1}$ via ~30 m of ¼" (6.35 mm) outer diameter (OD) perfluoroalkoxy (PFA) tubing maintained at 55ºC, which was then subsampled by the instrument through ~100 cm of 1/16" (1.59 mm) OD polyetheretherketone (PEEK) tubing maintained at 60ºC. VOCs with proton affinities higher than that of water (>165.2 kcal mol$^{-1}$) were ionized via proton-transfer reaction utilizing $H_3O^+$ as primary ions, then subsequently separated and detected by a ToF-MS with mass resolving power of ~4000 amu/Δamu. Ions were measured from $m/z$ 17-400 every two minutes. Residence time from the sample inlet on the 4 m tower to the drift tube was less than 5 s. Instrument backgrounds were quantified roughly every five hours for 20 minutes by measuring VOC-free air generated by passing ambient air through a heated catalytic converter (375ºC, platinum beads, 1wt% Pt: Sigma Aldrich). Calibrations were performed every four days, via dynamic dilution of gas standard mixtures containing 25 individual VOCs (stated accuracy 5% at ~1 ppmv; Apel-Riemer Environmental, Inc., Miami, FL; Permar et al., 2021) with overall uncertainty <15% (Table S1). Formaldehyde was calibrated post campaign with a certified standard via the method above, and humidity dependence was also accounted for, leading to higher uncertainty (40%). Formic acid and acetic acids were calibrated with a permeation device deployed in the field, and have uncertainties of ~30% (Table S1, Permar et al., 2021). Instrument sensitivities for all remaining VOCs that are not directly calibrated were estimated theoretically based on their molecular dipole moment, polarizability, functional groups (Sekimoto et al., 2017), and following procedures developed in our previous field campaign (Permar et al., 2021). The overall uncertainty for this method is estimated to be 50% for most species, consistent with previous work (Table S1, Sekimoto et al., 2017; Permar et al., 2021).

Peak fitting and integration were performed with the PTR-MS Viewer 3.2.12 post processing software (Ionicon Analytik GmbH, Innsbruck, Austria). Molecular formulae and compound names were assigned utilizing the workflow published in Fig.

S1 of Millet et al., (2018), and based on comparison with previously published PTR-MS libraries (Permar et al., 2021; Pagonis et al., 2019; Koss et al., 2018). The limit of detection (LOD) for each species was defined as two times the standard deviation ($\sigma$) of instrument blank or zero values. Species with LOD larger than the 95th percentile of measured ambient values were removed from the analysis (~50 of 126 ions removed, collective contribution <5% of instrument signal). Wind, $NO_x$, and $C_6$-$C_8$ aromatic VOC measurements were used to filter local anthropogenic influence from camp activities. Specifically, we removed data points that were simultaneously associated with the direction of the camp (15º to 60º NW), low wind speed associated with stagnant conditions (<1.5 m/s), high $NO_x$ (>0.5 ppbv, 95[th] percentile), and high individual anthropogenic VOC abundance ($C_6$-$C_8$ aromatics>0.6 ppbv, 95[th] percentile). This removed approximately 15% of measurements. All viable 75 VOC species/masses measured by PTR-ToF-MS and their measurement statistics are listed in Table S1.

**2.4 Ancillary measurements**

$NO_x$ (sum of NO and $NO_2$) was measured using a custom-built high sensitivity (~ 5 pptv detection limit) single channel chemiluminescence analyzer as described by Fontijn et al. (1970), that monitors $NO_x$ in ambient air using a photolytic converter and automated switching valves to alternate between NO and $NO_2$ modes every 30 minutes. Calibration was completed once a day by dynamic dilution of a 1.5 ppmv compressed NO gas standard (Scott-Marrin, Inc. Riverside, CA, USA). $O_3$ was measured using an ultraviolet (UV) absorption monitor (TEI model 49C, Thermo Fisher Scientific, MA, USA). The instrument underwent automated daily zero and span checks and was calibrated before and after the field campaign against a TEI model 49C primary standard calibrator. Overall uncertainty in $O_3$ measurements is estimated to be ±1 ppbv for 10-minute averaged data. Gas chromatography and mass spectrometer with flame ionization detection (GC-MS/FID) was utilized to measure a select number of hydrocarbons, including butane, pentane, and isohexane. These measurements are discussed in more detail in section 3.3.2. For a full description of the GC-MS/FID technique, see Angot et al., 2020.

**2.5 GEOS-Chem chemical transport model**

We applied a nested grid version of the GEOS-Chem chemical transport model to simulate VOC mixing ratios at TFS (version 13.3.2; https://doi.org/10.5281/zenodo.5711194; Bey et al. 2001). In this study, we implemented a custom nested grid centered over Alaska ranging from 50°N to 75°N and 130°W to 170°W, with $93 \times 128$ grid cells at 0.25 ° × 0.3125° (latitude × longitude) and 47 vertical layers (Kim et al., 2015; Wang et al., 2004). The model is driven by NASA GMAO GEOS-FP assimilated meteorological data and is run with timesteps of 5 minutes for chemistry and transport, and 10 minutes for emission and deposition. Chemical boundary conditions were taken from a 4º × 5º global simulation every 3 hours. Model spin-up for initialization employed a two-year simulation at the global 4º × 5º resolution followed by 1 month at the nested domain prior to the study period. Emissions were computed using the HEMCO module (Keller et al., 2014), using the Community Emission Data System (CEDS) for anthropogenic emissions (McDuffie et al., 2020; Hoesly et al., 2018), and the Global Fire Assimilation System (GFAS) for biomass burning emissions (Kaiser et al., 2012).

The Model of Emissions of Gases and Aerosols from Nature (MEGANv2.1) within GEOS-Chem implemented by Hu et al. (2015) was used to calculate biogenic VOC emissions (Guenther et al., 2012). Average monthly biogenic emissions of isoprene, methanol, and acetone for the model Alaska domain during June 2019 are shown in Figure 2. MEGANv2.1 computes biogenic emissions for each model grid cell based on the fractional coverage of 15 plant functional types (PFTs) and the corresponding base emission factor for each VOC under standard conditions. PFT distributions from the Community Land Model version 4 (CLM4) (Lawrence et al., 2011) within ~50 km radius of TFS include broadleaf deciduous boreal shrub (56%), bare land (34%), and Arctic C3 grasses (7%), with minimal (<3% total) contributions from other PFTs (Fig. S1) (Guenther et al., 2012). The MEGANv2.1 base emission factor for isoprene is 4000 $\mu$g m$^{-2}$ h$^{-1}$ for broadleaf deciduous boreal shrub but just 1600 $\mu$g m$^{-2}$ h$^{-1}$ for Arctic C3 grass, resulting in large predicted isoprene emission gradients in the Alaskan North Slope region. MEGANv2.1 accounts for the major environmental processes driving emission variations, including light, temperature, leaf age, leaf area index, and $CO_2$ inhibition.

Later, we evaluate the temperature (and light) dependence used to drive biogenic emissions in MEGAN. For isoprene, emissions are treated as 100% light-dependent, with temperature activity factor ($\gamma_T$) calculated as:

$$\gamma_T = E_{opt} \left[ 200 \frac{exp\ (C_{T1}x)}{200 - C_{T1}(1 - exp\ (200\ x))} \right] \tag{1}$$

where

$$x = \frac{\left[ \left( \frac{1}{T_{opt}} \right) - \frac{1}{T} \right]}{0.00831} \tag{1a}$$

$$T_{opt} = 313 + (0.6(T_{240} - 297) \tag{1b}$$

$$E_{opt} = C_{eo} \times exp\ exp\ \left( 0.08(T_{240} - 297) \right) \tag{1c}$$

In the above equations, $T$ is the 2m air temperature which is assumed to be equivalent to the leaf temperature, and $T_{240}$ is the average surface air temperature over the past 240 h. $C_{T1}$ and $C_{eo}$ are both VOC-dependent empirical coefficients, equal to 95, and 2, respectively for isoprene.

On the other hand, $\gamma_T$ for methanol is computed as a weighted average of a light-dependent fraction (80%) following eqn. 1 and a light-independent fraction (20%) following eqn. 2:

$$\gamma_T = exp\,[\beta(T-303)] \hspace{5cm} (2)$$

where β is an empirically determined coefficient (set equal to 0.08 for methanol, Guenther et al 2012).

Evaluation of temperature and light response within models on the effect of BVOC emissions in higher latitudes is crucial for addressing discrepancies in model simulations, as Arctic plants appear to respond to warming differently than plants from low latitudes (Rinnan et al., 2014). In addition to landscape changes in plant composition and functional type, tundra plants with relatively dark surfaces and low growth forms may also experience higher leaf temperature than air temperature measured at

245 heights (~2 m) provided by weather stations. Studies have observed large temperature oscillations among surface vegetation (10 to 26°C) , and differences of between 7-20°C when comparing air and surface temperatures (Seco et al., 2020; Lindwall et al., 2016). This could lead to larger emissions than anticipated in current models, and  identified challenges in accurately estimating BVOC emissions are thus closely related to having accurate estimations of temperature and PFTs, along with representation of long-term vegetation changes (Tang et al., 2016).

For comparison with observations, we sample the surface model grid cell over TFS on an hourly basis. CLM4 indicates that the vegetation distribution is relatively consistent over the spatial scale of the GEOS-Chem grid surrounding (~100 km) TFS. Plant survey data supports this (Fig. S1; Angot et al., 2020). Figure 1 shows the GEOS-FP meteorological inputs used to drive GEOS-Chem and MEGANv2.1 biogenic emissions. In general, simulated and observed temperatures agree to within ~3℃,

and PAR agrees to within 20%. Modelled hourly surface temperature was on average only 0.4°C higher than observed ambient temperature during peak PAR hours (10:00 to 15:00). However, simulated hourly temperature exhibited a larger deviation from observational "night time" values (±2.0°C) between 20:00 and 04:00, and when PAR was lower. We discuss how these discrepancies can affect biogenic VOC emission predictions in later sections.

**3 Results and discussion**

**3.1 Major VOCs in the Alaskan Arctic tundra**

We present measurements of 78 identified VOCs in this study, including 75 compounds measured by PTR-ToF-MS, and 3 complementary VOCs measured by GC-MS/FID that were not included as part of Angot et al., 2020, but were quantified and are useful in attributing anthropogenic sources of VOCs (butane, pentane, isohexane, Table S1). Among the 78 measured

species, eight major masses account for over 80% of the measured total carbon mass. These eight major VOCs include formaldehyde, methanol, acetaldehyde, formic acid, acetone, acetic acid, isoprene, and the sum of isoprene oxidation products methacrolein (MACR) and methyl vinyl ketone (MVK). We primarily focus on these species in this section due to their widespread global abundance and potential to significantly alter oxidative chemistry. Additionally, these species represent some of the most commonly globally studied VOCs to date, which allows us to compare our rare measurements from the

Arctic tundra to lower latitude ecosystems, as well as to evaluate our current understanding of VOC emissions within CTMs.

In later sections, we examine the measured total VOCs and their role in OH reactivity and ROC. Table 1 lists measurement statistics for the eight major VOCs mentioned. Figure 3 shows the time series of hourly-average ambient mixing ratios and corresponding GOES-Chem outputs. For the first four weeks of the field campaign, all VOCs remained at relatively low levels, reflecting cooler daily average air temperatures (7.4 ± 2.6°C) that occasionally dropped to freezing and limited biological activity. During the last few days of the study (June 19 to June 22), rising daily average temperatures (13.7 ± 3.2°C) led to a threefold enhancement in the abundance of several BVOCs relative to their campaign average.

The most important terpenoid BVOC, isoprene, and the sum of its oxidation products MACR+MVK reached hourly maximum values of 0.54 ppbv and 0.45 ppbv, respectively, near the end of the campaign when air temperatures were highest (>20°C). These maximum values are roughly an order of magnitude higher than the corresponding campaign mean values (i.e., isoprene 0.06 ± 0.06 ppbv and MVK+MACR 0.06 ± 0.06 ppbv; mean ± 1σ; Table 1), and these values are consistent with ambient measurements from GC-MS/FID measurements to within 10% (Angot et al., 2020). Our observations also appeared to capture the beginning of the isoprene seasonal cycle for the Alaskan Arctic tundra. The onset of isoprene emissions near TFS is about one month later than in mid-latitude ecosystems, reflecting the seasonal and latitudinal gradient in plant phenology (i.e., late May or early June in midwestern, northeastern, or southeastern US; McGlynn et al., 2021; Hu et al., 2015; Goldstein et al., 1998). Additionally, it is well known that the capacity for leaf-level isoprene emissions is delayed developmentally, with leaves becoming photosynthetically active weeks before isoprene emission begins. This delay is significantly affected by growth temperature, and the air temperature of previous days to weeks can affect the basal rate of isoprene emissions (Sharkey et al., 2008). As shown here, a rapid ~10-fold enhancement in isoprene concentrations was observed within just a few weeks. Our observed maximum isoprene mixing ratio is roughly a factor of three lower than previous measurements at a nearby site (i.e., hourly mean up to 1.5 ppbv; Potosnak et al., 2013), likely due to seasonal variation. Elevated isoprene abundance was primarily associated with northern and southerly wind directions.

As with observations in other ecosystems, isoprene and MACR+MVK measured at TFS were well correlated with each other ($r^2$>0.75). Concentrations of MACR+MVK showed a diurnal pattern similar to that of PAR and temperature, highlighting biogenic sources (Fig. 5). The ratio between isoprene and MACR+MVK depends upon several factors, including atmospheric mixing, distance from isoprene emitters, and local oxidant chemistry, which hinges on the concentration of $NO_x$ (Hu et al., 2015; Apel et al., 2002; Stroud et al., 2001). The average hourly isoprene/MACR+MVK ratio was ~1 and decreased slightly during the enhancements observed at the end of the campaign (0.9), likely due to enhanced photochemistry. Lower-latitude studies investigating the isoprene/MACR+MVK ratio suggest that values ≥ 1 indicate an approximate transport time less than one isoprene lifetime, with values less than 0.5 indicating more regional aged emissions (Hu et al., 2015). Isoprene lifetimes, modulated by OH abundance, are estimated to be <1 h at lower latitudes based on typical OH concentrations (~1× $10^6$ molecules $cm^{-3}$) (Wells et al., 2020; Hu et al., 2015; Warneke et al., 2004). The 24h median OH concentration simulated by GEOS-Chem during this period (7.8 × $10^4$ molecules $cm^{-3}$) implies an isoprene lifetime of approximately 3.6 hrs in the area

around TFS. Based on this lifetime and the average daytime (08:00 to 20:00) wind speed of roughly 3.5 m s$^{-1}$, this would indicate an average transport range of roughly 50 km, an area whose PFT is mostly broadleaf deciduous boreal shrubs according to CLM4 land cover (Section 2.5, Fig. S1).

Of the major OVOCs listed in Table 1, methanol showed the highest mean mixing ratio (3.1 ± 1.5 ppbv), followed by acetone
(1.1 ± 0.31 ppbv), formaldehyde (0.84 ± 0.2 ppbv), formic acid (0.50 ± 0.63 ppbv), acetic acid (0.28 ± 0.39 ppbv), and acetaldehyde (0.24 ± 0.15 ppbv). During the ATom aircraft mission, ~0.70-1.40 ppbv of methanol (25th -75th percentile range) were observed in the Arctic boundary layer during summer 2016 (Bates et al., 2021), but higher levels were measured in the free troposphere (~2.50 ppbv). The mean mixing ratio of acetone reported in this study is comparable to that measured at Utqiagvik, AK, during the OASIS-2009 field campaign in March-April 2009 (0.90 ± 0.30 ppbv, Hornbrook et al., 2016), but
roughly 75% higher than the mean mixing ratio reported in Pernov et al., 2021 from measurements at Villum Research Station in Greenland (0.61 ppbv) between April and October.

Highly variable mixing ratios of formic and acetic acid that are 3-5 times higher than those observed at Toolik (formic acid 1.23 ± 0.63 ppbv, acetic acid 1.13 ± 1.54 ppbv; Mungall et al., 2018) were observed under diverse environmental conditions
(cold, cloudy and warm, sunny) during early summer near the ocean in Alert, Nunavut, Canada. However, Pernov et al., 2021 reported measurements (with 1σ in parenthesis) of formic (0.45 ± 0.37 ppbv) and acetic acid (0.20 ± 0.15 ppbv) in Greenland that are in closer agreement to our observed values. Simulated annual average global distributions of acetaldehyde mixing ratios suggest there is between 50-200 pptv of acetaldehyde in the Alaskan arctic tundra between the boundary layer and mid troposphere (Millet et al., 2010), with the highest mixing ratios correlated to high biogenic emissions and precursor alkenes.
This range is within the variability of the average value of acetaldehyde measured at TFS.

Enhancements of all major OVOCs in Toolik tended to be strongest in air flow from both the north and south (Fig. 4), and correlated with elevated isoprene. Given the low wind speed and low abundance in anthropogenic tracers such as aromatic compounds, it is unlikely that measured OVOCs were chemically produced from precursor alkenes that may have been emitted
from Prudhoe Bay to the northeast (~200 km away). It also is unlikely that any significant OVOC enhancements observed during this campaign were due to biomass burning for several reasons. First, wildfire detections within Alaska were minimal throughout the duration of the campaign (May to June 2019) and located primarily south of the Brooks range, according to a global biomass burning emission inventory and satellite remote sensing of formaldehyde (Zhao et al., 2022). The abundance of formic and acetic acid can also be indicative of whether wildfire emissions impacted our dataset. For instance, studies have
long shown significant secondary production of organic acids in wildfire plumes, with acetic:formic acid ratios >> 1 (Akagi et al., 2011; Trentmann et al., 2005; Yokelson et al., 2003). We observed formic acid abundance roughly twice that of acetic acid throughout the campaign, which is inconsistent with biomass burning as a significant source. Additionally, though maleic anhydride, a secondary VOC formed from rapid oxidation of smoke and a marker for aged biomass burning (Coggon et al.,

2019) exhibited a large enhancement of 30-60 pptv at the end of the campaign, this enhancement only lasted for <10 hours.
The rest of the monitoring period, maleic anhydride was close to, or below the limit of detection (~5-10 pptv). Finally, model simulations comparing OVOC abundance with and without the inclusion of biomass burning emissions show negligible (<5%) differences in simulated OVOCs within this domain (Fig. S2), again reflecting minimal wildfire activities during the campaign period. For these reasons, we believe that biomass burning was not a significant contributor of the measured VOCs throughout the field campaign.


### 3.2 GEOS-Chem+MEGANv2.1 simulated major VOCs

Table 2 lists comparisons of major observed species to the corresponding predictions from GEOS-Chem. Observation:model comparisons indicate good agreement within ~10% for both isoprene and MACR+MVK. Good model:measurement correlation is obtained for these species throughout the campaign ($r^2$>0.6). The simulated hourly isoprene/MACR+MVK ratio
(1.24 ± 0.03) is within 15% of the observed value (1.07 ± 0.03), showing that fresh emissions without extensive chemical processing are accurately captured in the model (Fig. S3). The model is also generally able to capture the $NO_x$ levels at TFS, which on average were measured to be 0.10 ± 0.07 ppbv throughout the campaign, and simulated at 0.15 ± 0.10 ppbv, reflecting a low $NO_x$ environment.

We found that the overall simulated temperature activity factor ($\gamma_T$) for isoprene is underestimated by approximately 20% for both campaign-mean observed $\gamma_T$, and during daytime only values (08:00 to 20:00) (Fig. S4), yet the model can reproduce observed isoprene abundance to within 10%. $\gamma_T$ was enhanced by a factor of ~2.5 at the end of the campaign relative to the rest of the monitoring period, which supports the idea that increased biogenic activity was primarily responsible for the VOC enhancements observed towards the end of the campaign and reinforces the notion that wildfires were not a significant source
of these enhancements. We also derive $\beta$ coefficients for isoprene and methanol to determine the temperature response of emissions, with higher beta indicating a steeper temperature response curve and vice versa. Isoprene and methanol both exhibit light dependence, thus we controlled for this by only looking at $\gamma_T$ during daytime hours (08:00 to 20:00) when PAR was >400 $\mu mol\ m^{-2}\ s^{-1}$. However, we find that the simulated (0.114; 95% CI: 0.09–0.138) and observed (0.161; 95% CI: 0.149–0.173) $\beta$ coefficients for isoprene (Fig. 6a) are not statistically consistent with one another. Here, $\beta$ indicates that simulated isoprene
mixing ratios are less sensitive to assimilated temperature compared to the observed relationship, particularly when ambient temperatures are higher than ~10ºC, thereby implying that the response to temperature should be steeper. However, this may also be partially due to differences in observed versus assimilated meteorology during some of the warmest days. Additionally, short-lived species would be very sensitive to any model errors in the mixing height, and the $\beta$ inconsistency found here could suggest model errors in emissions and/or mixing. CTMs tend to have difficulties simulating the shallow night-time mixing
layer and its evolution, and a small discrepancy could result in large errors for the calculation of atmospheric concentrations. We utilize balloon data reported in Angot et al., 2020 to evaluate the vertical mixing dynamics within GEOS-Chem. Figure S5 shows vertical profile and mixing data of ambient isoprene concentrations measured by a tethered balloon between 15 July

2019 and 16 July 2019 (see Angot et al., 2020 for full description of methods), and concentrations simulated by GEOS-Chem for the bottom three layers (0-350 m above ground level). Observations show isoprene to be well-mixed between 0-250 m during the day, which the model is generally consistent with. However, at night (21:00 to 6:00 local time) concentrations of isoprene become more stratified, which is challenging for the model to capture.

On some days, observations of PAR are overestimated, while in other instances PAR is underestimated (Fig 1b), leading to imperfect agreement between observed and simulated PAR (slope = $1.22 \pm 0.03$ ; $r^2 = 0.63$). In a situation where $\gamma_P$ is overestimated but $\gamma_T$ is underestimated or vice-versa, the error in the activity responses might offset one another resulting in no difference between observed and simulated isoprene abundance. We controlled for this by only looking at $\gamma_T$ during daytime hours (08:00 to 20:00) when PAR was >400 µmol m$^{-2}$ s $^{-1}$. We find that despite the errors in assimilated environmental variables (T, PAR) leading to ~20% underestimation in $\gamma_T$, isoprene is only slightly (~10%) overestimated by the model (Fig. 3, 5). However, MACR+MVK is a more robust tracer to evaluate model isoprene emission due to its longer lifetime and decreased sensitivity in model errors due to vertical mixing, OH chemistry, or plant functional type (Hu et al., 2015). Given that the errors caused by assimilated temperature and PAR inputs are minimal, we conclude that GEOS-Chem+MEGANv.2.1 can reproduce regional isoprene emissions to ±20%, constrained by our observations at TFS. However, we note that our results are limited to the early growing season, and may also be variable in later months (July, August) due to large discrepancies between surface and air temperatures (Seco et al., 2020; Lindwall et al., 2020). Nonetheless, better meteorological inputs can help further improve the prediction of isoprene emissions.

Further comparisons of measured versus simulated OVOC abundance shown in Figs. 3 and 5 yield varying results. Simulations of acetone and acetaldehyde abundance were both underestimated by ~20-30% but within the combined variability of measurements and model representation errors, suggesting an overall good understanding of their budgets in the remote Arctic tundra. Some of the most striking differences are the significant model underestimations for methanol, formaldehyde, formic acid, and acetic acid. GEOS-Chem systematically underestimates observed methanol by a factor of almost four but is substantially correlated with observations ($r^2 = 0.57$). The recently identified secondary production of methanol from $CH_3O_2$ + OH and self-reaction of $CH_3O_2$ is incorporated in the model version used in this study, and these reactions have been suggested to account for ~30% of global methanol sources (Bates et al., 2021). However, including these reactions is insufficient in capturing the observed methanol level at TFS. Biogenic methanol emissions increase exponentially with temperature (Guenther et al., 2012), thus evaluating the temperature dependence will allow us to investigate if there is any model bias within this relationship that could explain the underestimated methanol abundance. Figure 6b shows ambient methanol mixing ratios versus temperature for both observations and simulations and the exponential fits following Equation 2. The two derived $\beta$-coefficients are statistically consistent with one another, with 95% confidence intervals of 0.104—0.136 (observations) versus 0.097—0.123 (simulation). Such agreement implies the model biogenic temperature response is not a significant contributor to the model:observation discrepancy. We further carry out a sensitivity test with tripled biogenic

methanol emissions in the Alaskan domain (Fig. S2). This leads to a significant model improvement (model bias ~10%; $r^2$ = 0.6). Thus, the above analyses suggest that the negative bias in the base model is due to MEGANv2.1 underestimating biogenic methanol emissions in Alaska by nearly 200%. There appears to be no wind direction bias in comparison between observed and simulated mixing ratio for methanol or for any of the major eight VOC species mentioned here (Fig. S6). Thus, we infer that the base emission factors for methanol in the corresponding relevant PFTs are too low in MEGANv2.1 (i.e., default 500-900 µg m$^{-2}$ h$^{-1}$ recommended values for needleleaf evergreen boreal tree, broadleaf deciduous boreal shrub, and Arctic C3 grass which together account for >80% of land area in Alaska according to the PFT distribution in CLM4; Figure S1; Figure 2).

GEOS-Chem underestimates formaldehyde concentrations by more than a factor of three (Figures 3 and 5; Table 2). Such underestimation is likely also compounded by some PTR-ToF-MS measurement uncertainty associated with varying ambient humidity and the low proton affinity of formaldehyde (± 40%; Table S1; Permar et al., 2021), but this alone is not enough to explain the large model and observation discrepancy. Though methanol oxidation can be a source of formaldehyde (Hu et al., 2011), our sensitivity test with tripled biogenic methanol emission only leads to an average increase of 0.03 ppbv (or ~3%) in simulated formaldehyde compared to the base simulation, reflecting slow atmospheric oxidation due to cool temperatures, low NO$_x$ and low OH in the Arctic environment. A recent study of boreal environments in Alaska suggests that formaldehyde vertical column densities observed from space are primarily driven by background methane oxidation and primary emissions from wildfires when available, rather than a biogenic source (Zhao et al. 2022). However, our additional sensitivity run with global biomass burning emissions turned off contributes to less than 5% change among modelled VOCs including formaldehyde, and does not affect any of the observed species enhancement at the end of the campaign (Fig. S2). In addition, observed formaldehyde shows exponential increases towards enhanced air temperature ($r^2$=0.5, Figure S7), likely indicating the biogenic origin of its precursors, and pointing to the systematic model underprediction of secondary formaldehyde production. Indeed, the strong diurnal cycle for formaldehyde shown in Fig. 5 compared to almost no diurnal cycle in the model suggests that missing VOCs, or missing direct sources of formaldehyde must be significantly responsible for the discrepancy, rather than methane oxidation alone.

CTM and box model comparisons of formic and acetic acid to observations have been shown to persistently underestimate their mixing ratios, particularly in Arctic and northern mid-latitude environments (Schoebesberger et al., 2016; Stavrakou et al., 2012; Paulot et al., 2011). Indeed, GEOS-Chem underestimates both formic and acetic acid at TFS by a factor of over 12 and 8.5, respectively (Table 2). Additionally, neither compound had observations that were well correlated with model simulations ($r^2$<0.2). These results highlight the complexity and variability associated with formic and acetic acid and imply that current CTMs have an incomplete understanding in sources and chemistry associated with these compounds. The uncertainty associated with simulating these organic acids is likely also compounded by uncertainties in formaldehyde and

methanol emissions, as these species are interconnected through several photochemical pathways that are not included in the GEOS-Chem version used here (Franco et al., 2021).

### 3.3 Reactive organic carbon (ROC) from measured and modelled VOC species

ROC are critical in the formation of secondary species and contextualizing atmospheric processes, but our understanding of their abundance, budget, and chemical impact has not been probed in Arctic environments. North American studies of ROC in mid-latitude forests and urban areas suggest alkanes can account for anywhere between 15-30% of observed ROC by mass, with organic aerosol accounting for another 3-17% (Heald et al., 2020; Hunter et al., 2017; Heald et al., 2008). These species were not extensively measured at TFS and thus our measurements of ROC in this area should be taken as lower limits. However, prior work has shown that the species that were measured are expected to account for the majority of ROC and OHr (e.g., Fig. 2 in Hunter et al., 2017). Thus despite some limitations, in the following sections we present one of the most comprehensive ROC and OHr assessments to date for the Arctic tundra region, utilizing data from the entire mass spectrum of PTR-ToF-MS measurements and complementary GC-MS/FID data. This information will help to probe if any significant amount of missing "unknown" ROC exists within the Arctic atmosphere, and to what extent "known" compounds contribute to overall ROC abundance. We also evaluate GEOS-Chem to test if current models miss a significant amount of reactive carbon or reactivity in this remote atmosphere. Figure 7 shows the full mass spectrum of PTR-ToF-MS measurements at TFS, as a function of median species concentration (based on hourly data) measured throughout the campaign, versus mass to charge ratio (*m/z*). For simplification purposes, masses were generally subcategorized based on their structure and functional groups (Table S1).

For reasons discussed earlier, we do not attempt to segregate periods with potential wildfire influence in our TFS data set, and instead examine the overall campaign average. The total molar mixing ratio based on median VOC abundance (tVOC) measured by the PTR-ToF-MS was $6.29 \pm 0.36$ ppbv ($10.8 \pm 0.5$ ppbC; $5.3$ $\mu$gC sm$^{-3}$). Adding complementary GC-MS/FID butane, pentane, isohexane measurements only adds $0.04$ ppbv ($0.19$ ppbC, or $0.10$ $\mu$gCsm$^{-3}$), resulting in contributions less than 1% of the measured tVOC at TFS. Based on these and other anthropogenic tracers measured by PTR-ToF-MS, we conclude there was negligible influence of anthropogenic emission in the Alaskan tundra during the study period. The measured tVOC at TFS is considerably lower than the average from mid latitude forests ($26.7$ to $36.5$ $\mu$gCsm$^{-3}$, Heald et al., 2020; Hunter et al., 2017), urban environments ($4.0$ to $456$ $\mu$gCsm$^{-3}$, Heald et al., 2008), or biomass burning smoke ($148.3 \pm 29.6$ ppbv; $50$-$200$ $\mu$gCsm$^{-3}$, Permar et al., 2021). However, the tVOC measured at TFS is within the range of other remote areas ($4.0$ to $10$ $\mu$gCsm$^{-3}$) reported in Heald et al., 2008 from their cleanest sites. The largest contributors to molar tVOC mass (ppbv) were overwhelmingly dominated by OVOCs, including methanol (46%), acetone (17%), and formaldehyde (12%). Notable contributions also include formic and acetic acid, which together contribute an additional 8%, as well as acetaldehyde (3%) and ethanol (3%). These seven OVOCs represent almost 90% of the molar tVOC mixing ratio measured by PTR-ToF-MS. Isoprene had a negligible contribution (0.5%) by comparison, but we also note that our results only capture the early part

of the growing season and isoprene may therefore have a larger contribution than seen here. The remaining ~10% of molar tVOC mass was also mostly dominated by OVOCs, with minor contributions from N-containing species.

ROC mass concentrations were also dominated by OVOCs and account for over 80% of the total ROC carbon mass (Fig. 8a). In particular, acetone (1.59 $\mu gCsm^{-3}$; 3.25 ppbC), methanol (1.41 $\mu gCsm^{-3}$; 2.88 ppbC), and formaldehyde (0.36 $\mu gCsm^{-3}$; 0.74 ppbC) contribute to two thirds of the median ROC mass concentration measured. Lower latitude studies from southeast US forests have found that isoprene can account for almost a quarter of the observed ROC (Heald et al., 2020). Here, we find that isoprene only accounts for ~1.5% of the measured ROC mass at TFS. ROC mass based on all VOCs simulated by GEOS-Chem was 4.83 $\mu gCsm^{-3}$ (9.8ppbC), with sizable contributions from acetone, ethane, and lumped C4 alkanes (Fig. 8b). Though this absolute value agrees within 10% of the average conditions during the TFS campaign, the composition and distribution among individual species is variable and points to a larger discrepancy among observed and modelled ROC. For example, ethane and lumped C4 alkanes account for over a third of simulated ROC, but neither of these species could be confidently quantified by the PTR-ToF-MS at TFS aside from butane (part of ≥C4 alkanes). Assuming model estimates of ethane and the rest of ≥C4 alkanes are correct, this would account for an additional 1.8 $\mu gCsm^{-3}$ (3.5ppbC), or 7.00 $\mu gCsm^{-3}$ (14.3 ppbC) total (Fig. 8c). Interestingly, the isoprene contribution to ROC was similar (within 1%) in both observed and modeled estimates, but should be further verified with measurements from later in the growing season (July, August) where there is more discrepancy between surface and air temperatures used to derive isoprene emissions. The results shown here suggest that differences among known (e.g., methanol, formaldehyde) or unmeasured (e.g., alkanes) species are thus significant contributors to uncertainty in measured versus modelled ROC. As a result, future studies and comparisons of ROC in this environment would highly benefit from inclusion of alkane and aerosol measurements in addition to other terpenoid species particularly because of their propensity to be potential OA precursors.

## 3.4 Calculated OH reactivity (OHr) from measured and modeled VOCs

The calculated total OHr from VOCs is the sum of OH reactivity for each species $X_i$, which is the product of the OH reaction rate constant for each species $k_{OH+Xi}$ and its concentration $[X_i]$. Here we use the median mixing ratios throughout the campaign in the calculation to reflect the OHr general conditions observed at TFS and simulated in that area. Figure 9 shows individual contributions to calculated OHr from observations and GEOS-Chem simulations. Total calculated OHr based on median VOC concentration at TFS was 0.7 $s^{-1}$, which is ~5% of the OHr from VOCs measured during the 2013 SOAS campaign from forested areas in the southern US (~15 $s^{-1}$) (Heald et al., 2020). This result is also approximately an order of magnitude lower than the OHr due to VOCs from the 2010 CalNex campaign that took place in a more urban environment, and from mid-latitude ponderosa pine forests (~7 $s^{-1}$) (Heald et al., 2020, Hunter et al., 2017). Other studies from various forest environments have found OHr to be on the range of 1–42 $s^{-1}$ for mixed deciduous forests (Hansen et al., 2014), 8–25 $s^{-1}$ for coniferous forests (Mao et al., 2012), and 3–31 $s^{-1}$ for boreal environments (Praplan et al., 2019; Nölscher et al., 2012; Sinha et al., 2010) due to the higher abundance of isoprene or monoterpenes. Simulations of OHr from Safieddine et al., 2017 estimate reactivities of

0.8–1 s$^{-1}$, 3–14 s$^{-1}$, and 12–34 s$^{-1}$, over select regions in the remote ocean, continental mid latitudes, and tropics respectively, with the remote ocean estimate most comparable to our estimates in a remote area in the Alaskan arctic.

Safieddine et al. (2017) show that global mean estimates of OHr are dominated by aldehydes and isoprene, with isoprene accounting for anywhere between 3 to over 50% of the total OHr burden. Figure 9a shows the largest contribution to calculated OHr in the Alaskan Arctic tundra came from formaldehyde (0.17 s$^{-1}$) , isoprene (0.08 s$^{-1}$) , and acetaldehyde (0.08 s$^{-1}$) (together almost 50% of OHr). Terpenoid species including monoterpenes (0.06 s$^{-1}$) and sesquiterpenes (0.02 s$^{-1}$) make up a little over

10% of OHr. Though these terpene species account for an insignificant fraction of ROC, they contribute disproportionately to calculated OHr, highlighting their reactivity and importance.

Calculated model OHr due to VOCs is 0.5 s$^{-1}$ during the campaign. Modelled OHr is dominated by isoprene (0.15 s$^{-1}$) and monoterpenes (0.08 s$^{-1}$), which account for almost 50% of the total modelled value (Fig. 9b). Concentrations of total

monoterpenes were close to or below the detection limit in both PTR-ToF-MS and GS/MS techniques (2-20 pptv, Angot et al., 2020), but GEOS-Chem+MEGANv2.1 predicts them at levels similar to those at TFS (median of 0.02 ppbv) (Tab. 3). Contributions from acetaldehyde (0.07 s$^{-1}$) and formaldehyde (0.06 s$^{-1}$) account for another quarter of modelled OHr, with the remaining 14 VOCs responsible for the last ~25%. As with comparisons of ROC, the disparity among observed and modelled VOC OHr is largely due to underestimation in known compounds already included in the model (e.g formaldehyde), similar

to findings at lower latitudes (Millet et al., 2018). Unmodeled species are estimated to account for less than 5% of observed OHr.

The photochemical formation of ozone depends on the concentration of both NO$_x$ and total VOCs. Kirchner et al. (2001) proposed an indicator ($\theta$), as the ratio of OHr from NOx versus OHr from VOCs, to provide the sensitivity of potential ozone

formation in response to changes in concentration of VOC or NO$_x$.  When $\theta > 0.2$, ozone production is limited by VOC abundance (VOC-limited), and when $\theta < 0.01$, this implies a NO$_x$-limited regime and ozone production is insensitive to VOC concentration (Kirchner et al., 2001). Here, we utilize the average NO$_x$ mixing ratio from both observations (0.10 ppbv) and simulations (0.15 ppbv) to determine OHr from NO$_x$, then use it to derive $\theta$ by comparing to estimated VOC OHr. We find in this way a value of $\theta = 0.04$ from the observations compared to $\theta = 0.08$ from the model simulation. Both of these values

represent a transitional condition when ozone production is optimal and sensitive to any small perturbation, though observations point to somewhat higher NO$_x$ sensitivity. Both the observations and the simulations imply that moving to a VOC-limited regime would require a 2-5 fold increase in the amount of NO$_x$ given the current VOC abundance observed. Though this level of increase is unlikely, scenarios do anticipate shipping increases in the Arctic which are expected to increase concentrations of NO$_x$ (Gong et al., 2018; Eyring et al., 2005), resulting in predicted increases in Arctic surface ozone

concentrations (Granier et al., 2006; Brasseur et al., 2006). Arctic photochemistry could be further complicated by enhanced BVOCs due to warming temperatures or elevated VOCs from fire activities.

**4 Conclusions and implications**

Ambient PTR-ToF-MS and GC-MS/FID measurements of 78 VOCs in the Alaskan Arctic tundra show that OVOCs such as
methanol, acetone, and formaldehyde are the most abundant compounds present in this environment, and combined account
for nearly three quarters of the total observed VOC molar mass and over half of ROC. We find that GEOS-Chem can simulate
observed isoprene, MACR+MVK, acetone, and acetaldehyde to within the combined model and observation uncertainties
(±25%) with high correlation ($R^2>0.6$) during this early-season study period. However, we find threefold model
underestimation for formaldehyde and methanol, and roughly an order of magnitude underestimation in formic and acetic
acids, which likely affects the simulation of other species. These underestimations reflect significant knowledge gaps which
cannot be accounted for based on instrument measurement uncertainty alone. A sensitivity test that increased biogenic
methanol emissions by a factor of three resulted in model outputs that were in better agreement with observations, implying
that the base emission factors for methanol may be too low in MEGANv2.1 in the Arctic. Observed formaldehyde increases
exponentially towards higher air temperature, indicating its precursors are likely of biogenic origin and points to the systematic
model underprediction of its secondary production. We find that the temperature dependence of methanol emissions in
MEGANv2.1 is correct within the constraints provided by Toolik observations. The observed temperature dependence of
isoprene concentration was greater compared to simulations, for temperatures >10°C, likely reflecting model errors in
emissions and/or vertical mixing which warrants further investigation.

Calculated OHr from VOCs (0.7 s$^{-1}$) and ROC (5.3 µgCsm$^{-3}$) for the TFS area was only 5-10% of values seen in lower-latitude
forested and urban environments, reflecting the more 'pristine' and less chemically reactive nature of these high-latitude
environments. Supplementing unmeasured species with the simulated species (ethane, C5 or higher alkanes), we estimate 0.72
s$^{-1}$ OHr and 7.1 µgCsm$^{-3}$ ROC in Toolik, representing the most comprehensive estimate of VOC contributions to ROC and
calculated OHr in this area to date. Despite contributing <1% to total measured VOC mass, isoprene was responsible for 12%
of OHr, second only to formaldehyde, which accounted for 25% of the calculated OHr. Modelled OHr was primarily dominated
by isoprene and monoterpenes, together accounting for almost half of the total. Uncertainties in known species (methanol,
formaldehyde, organic acids) are some of the largest contributors to discrepancies between observations and our current
understanding within GEOS-Chem, highlighting the necessity for future targeted investigation of these compounds and their
sources in high latitudes.

The work presented here ultimately helps to bridge a significant gap in availability of observational reference data for this
ecosystem. Specifically, this study serves as a crucial evaluation of our knowledge of biogenic VOCs, ROC budgets, and OH
reactivity in high latitude environments, and represents a foundation for investigating and interpreting future changes in VOC
emissions as a result of climate warming in the Arctic. The extent to which the results of this point study can be extrapolated
beyond the Alaskan Arctic tundra will depend on surrounding PFTs and land cover as well as oxidative chemistry of the

environment. However, we expect the implications of this study to be broadly applicable given the widespread distribution of the PFTs surrounding TFS across the broader Arctic.

**Acknowledgements:**

This study was supported by U.S. National Science Foundation (# OPP1707569), a seed grant from the University of Montana University Grant Program (UGP), and NOAA Climate Program Office's Atmospheric Chemistry, Carbon Cycle, and Climate program (#NA20OAR4310296). D.K. was supported by the National Institute of General Medical Sciences of the National Institutes of Health (# P20GM103474). D.B.M. acknowledges support from NSF Grant #1932771. The authors would like to acknowledge high-performance computing resources and support from Cheyenne (doi:10.5065/D6RX99HX) provided by the

National Center for Atmospheric Research (NCAR) Computational and Information Systems Laboratory, sponsored by the NSF, and the U of Montana's Griz Shared Computing Cluster (GSCC). We thank CH2MHill Polar Services for the logistical support, and the Toolik Field Station staff for the tremendous assistance with the installation of the PTR-ToF. We also appreciate Bob Yokelson for the helpful discussions and Jacob Moss, Kaixin Cui, Katelyn McErlean, and Anssi Liikanen for assistance collecting the tethered balloon dataset used in this paper.

*Data availability*. Data are available upon request to the corresponding author.

*Author contributions.* DH, LH, AF, and DBM designed the experiments and acquired funding. CW acquired PTR-ToF-MS data during the field campaign, and DK contributed to post-processing PTR-ToF-MS data and data analysis. WP helped to

595 refine sampling technique and procedure. SC was responsible for initializing GEOS-Chem model runs and outputs. VS analysed the data and prepared the manuscript with contributions from all authors.

*Competing Interests.* The authors declare that they have no competing interests.

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

**Table 1**. Abundance (ppbv) of major volatile organic compounds (VOCs) measured at Toolik Field Station (TFS) in early summer 2019. Data have been filtered for stagnant air and local pollution influences from the field station (see text). An extended table containing the full mass spectrum of all identified PTR-ToF masses is provided in Table S1.

| Measured Mass (m/z) | Empirical Formula | Assigned Compound | Mean ± 1σ (ppbv)[1] | Median (ppbv) | Max (ppbv) |
|---|---|---|---|---|---|
| 31.018 | $(CH_2O)H^+$ | Formaldehyde | 0.84 ± 0.20 | 0.74 | 3.02 |
| 33.033 | $(CH_4O)H^+$ | Methanol | 3.13 ± 1.50 | 2.88 | 8.87 |
| 45.033 | $(C_2H_4O)H^+$ | Acetaldehyde | 0.25 ± 0.15 | 0.20 | 0.87 |
| 47.013 | $(CH_2O_2)H^+$ | Formic Acid | 0.50 ± 0.63 | 0.31 | 3.71 |
| 59.049 | $(C_3H_6O)H^+$ | Acetone | 1.11 ± 0.31 | 1.08 | 2.09 |
| 61.028 | $(C_2H_4O_2)H^+$ | Acetic Acid | 0.28 ± 0.40 | 0.17 | 2.20 |
| 69.070 | $(C_5H_8)H^+$ | Isoprene | 0.06 ± 0.08 | 0.03 | 0.54 |
| 71.049 | $(C_4H_6O)H^+$ | Methacrolein and Methyl Vinyl Ketone | 0.06 ± 0.08 | 0.03 | 0.45 |





**Table 2**. Comparisons and correlations of main observed VOCs hourly mixing ratios to hourly mixing ratios simulated by GEOS-Chem+MEGANv2.1, based on major axis regression.

| Compound | Slope (Observations/Simulations) | $r^2$ |
|---|---|---|
| Formaldehyde | $3.28 \pm 0.05$ | 0.30 |
| Methanol | $3.93 \pm 0.05$ | 0.57 |
| Acetaldehyde | $1.20 \pm 0.03$ | 0.11 |
| Formic Acid | $9.10 \pm 0.52$ | 0.04 |
| Acetone | $1.18 \pm 0.01$ | 0.55 |
| Acetic Acid | $10.4 \pm 0.50$ | 0.14 |
| Isoprene | $0.89 \pm 0.02$ | 0.63 |
| MACR+MVK | $1.10 \pm 0.03$ | 0.62 |




**Table 3.** Statistics of VOCs included in GEOS-Chem along with the corresponding observations at TFS. Blank entries for observed VOCs indicated that the VOC was either not detected by the PTR-ToF or was below detection limits.

| GEOS-Chem Species | Simulated median (ppbv) | Simulated IQR[a] | Observed median (ppbv) | Observed IQR[a] |
|---|---|---|---|---|
| Acetaldehyde | 0.19 | 0.14–0.25 | 0.20 | 0.15–0.30 |
| Acetic Acid | 0.02 | 0.03–0.07 | 0.17 | 0.09–0.30 |
| Acetone | 0.82 | 0.69–1.00 | 1.08 | 0.88–1.32 |
| Benzene | 0.06 | 0.04–0.13 | 0.02 | 0.01–0.03 |
| DMS | 0.01 | <0.01–0.02 | – | – |
| Ethane | 1.14 | 1.05–1.21 | – | – |
| Ethanol | 0.15 | 0.11–0.20 | 0.23 | 0.09–0.35 |
| Formaldehyde | 0.26 | 0.19–0.35 | 0.74 | 0.53–0.99 |
| Formic Acid | 0.05 | 0.03–0.11 | 0.31 | 0.16–0.58 |
| Isoprene | 0.06 | 0.04–0.10 | 0.03 | 0.02–0.07 |
| Lumped C4 Alkanes | 0.33 | 0.16–0.95 | – | – |
| MACR+MVK | 0.05 | 0.04–0.06 | 0.03 | 0.01–0.06 |
| MEK | 0.08 | 0.05–0.10 | 0.04 | 0.03–0.06 |
| Methanol | 0.7 | 0.52–0.96 | 2.88 | 1.98–4.03 |
| Monoterpenes | 0.02 | 0.01–0.04 | 0.014 | 0.01–0.02 |
| Propane | 0.12 | 0.07–0.13 | – | – |
| Toluene | 0.02 | 0.01–0.06 | 0.01 | <0.01–0.01 |
| Xylene | 0.01 | <0.01–0.04 | <0.01 | <0.01–0.01 |

[a]Interquartile range (IQR), which represents the 25th to 75th percentiles.

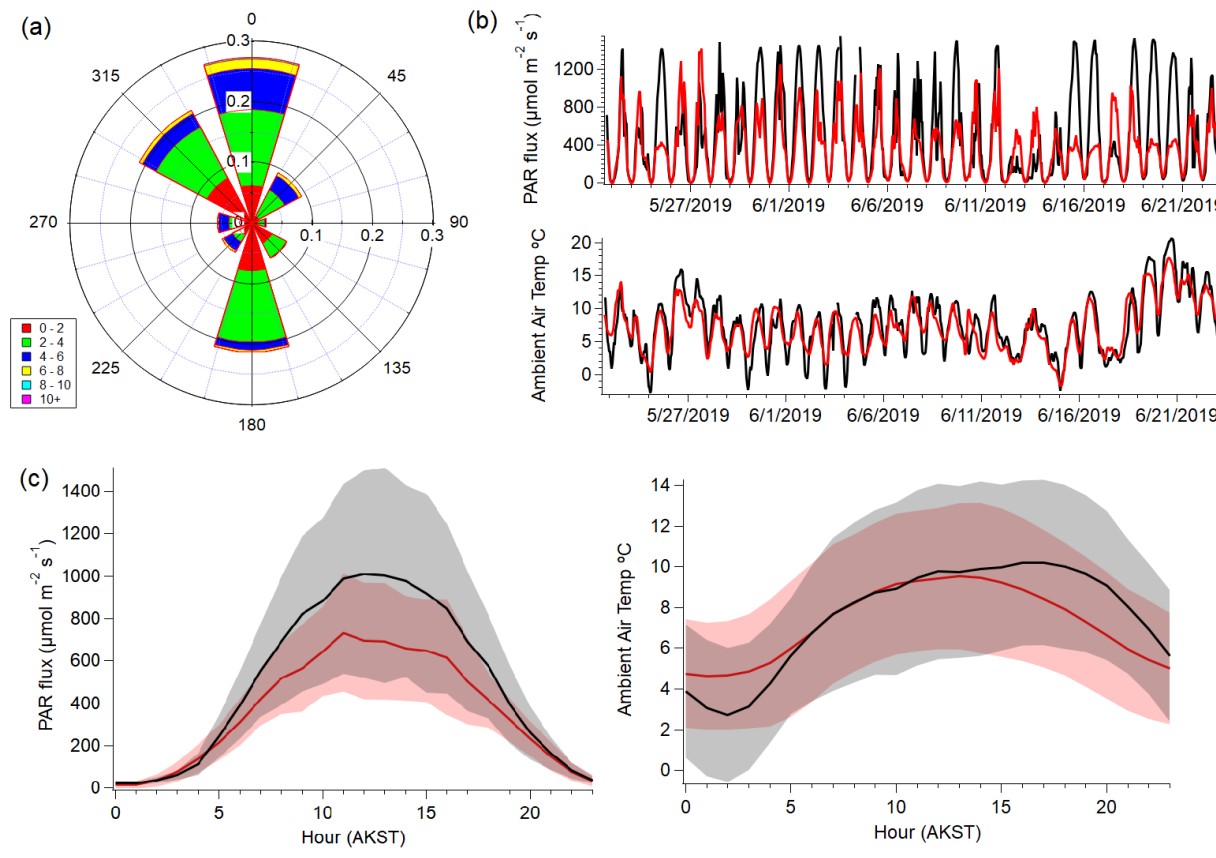

**Figure 1.** Meteorological data taken from TFS from May 22 to June 23: (a) probability (0-1) wind rose plot depicting wind direction and speed; (b) temporal traces of observed (black) and simulated (red) hourly photosynthetically active radiation (PAR) and surface air temperature (°C); (c and d) diel plots of observed (black) and simulated (red) PAR and temperature.
 Shaded areas represent one standard deviation (1σ).

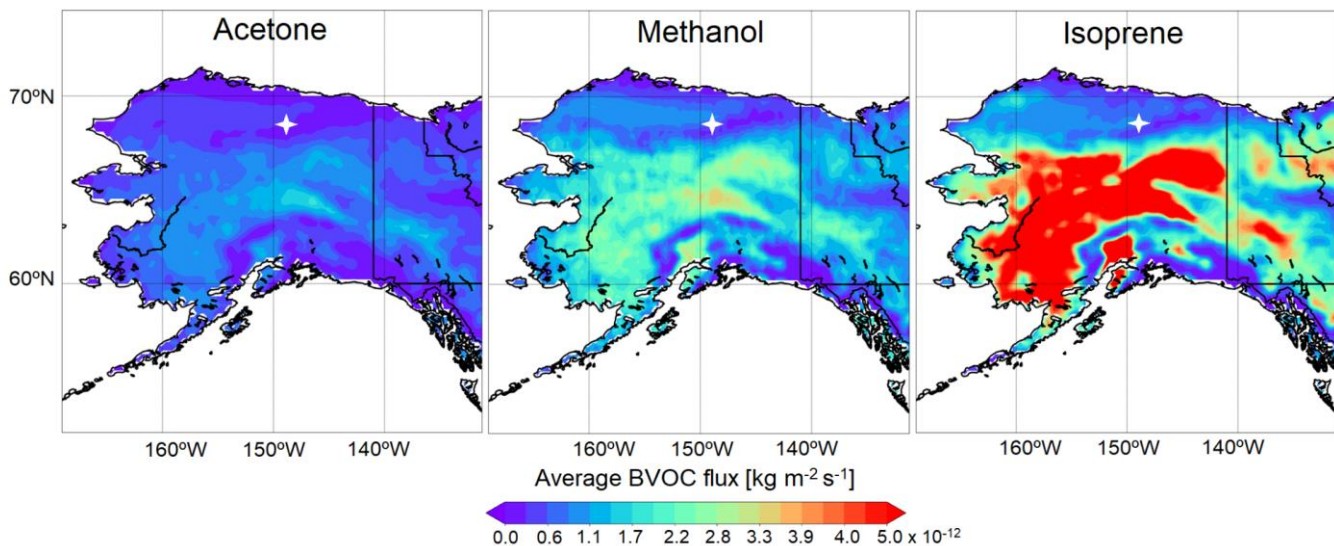

**Figure 2.** Monthly averaged biogenic emission estimates for acetone, methanol, and isoprene over the Alaska domain in June 2019, simulated using GEOS-Chem+MEGANv2.1. The location of Toolik is represented by the white marker.





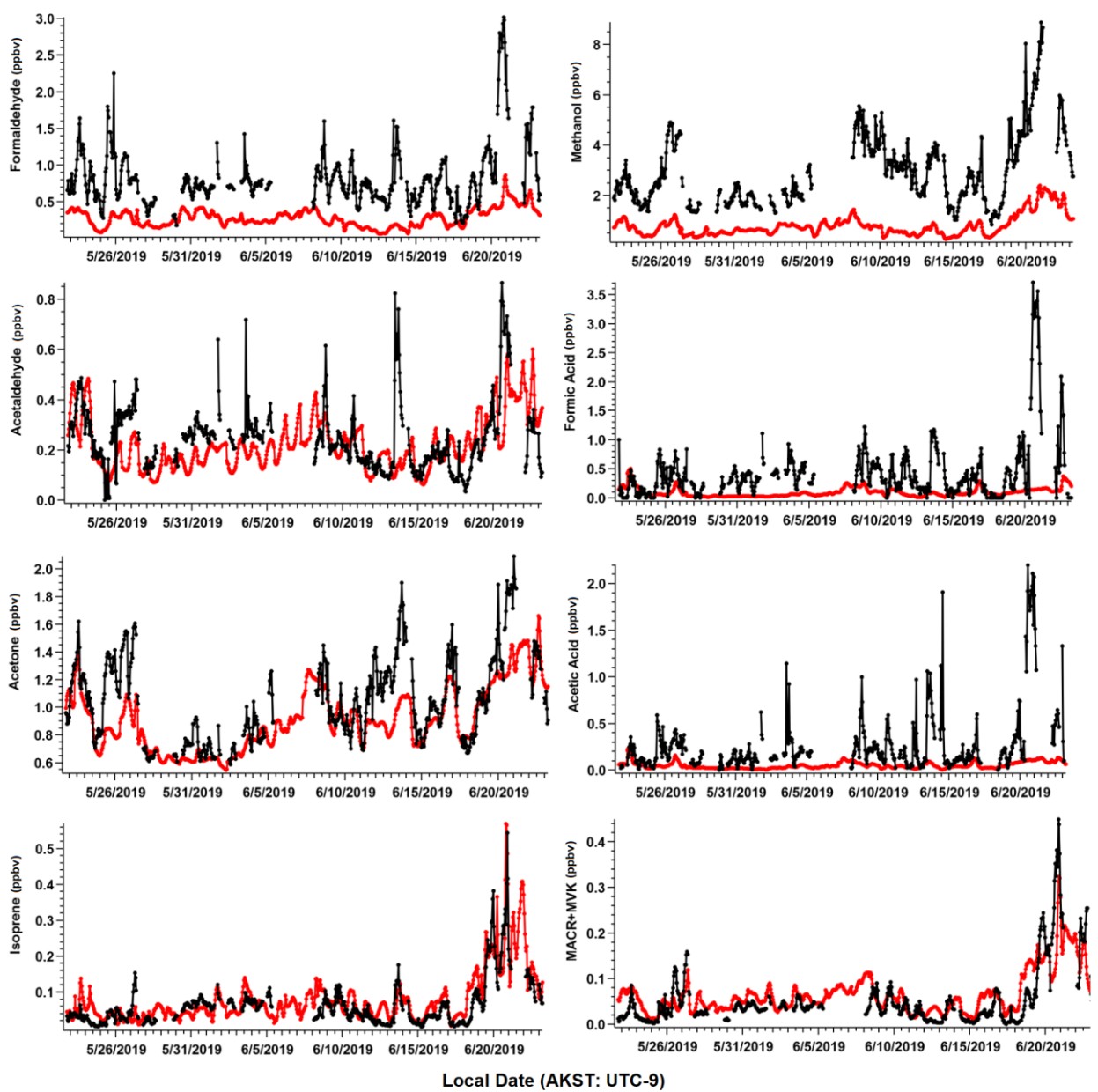

**Figure 3.** Ambient VOC mixing ratios (ppbv) as observed (black) and simulated by GEOS-Chem+MEGANv2.1 (red).
Observations shown are hourly averages computed from two-minute measurements and have been filtered for local pollution and stagnant air (see text).

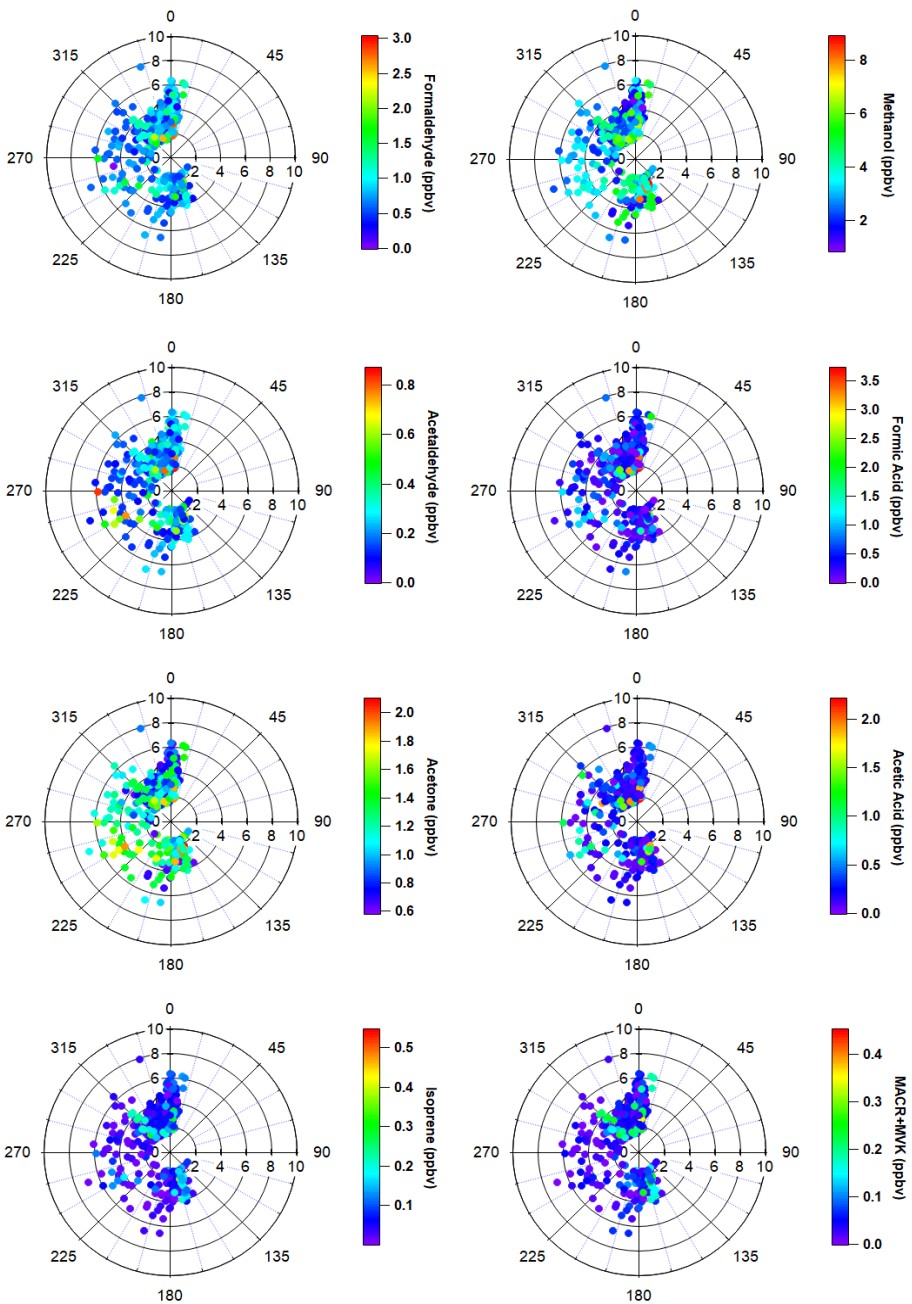

**Figure 4.** Polar wind plots of hourly wind speed, wind direction, and VOC mixing ratios (color scales, ppbv). Distance from the radius represents wind speed. Data have been filtered for local pollution and stagnant air.


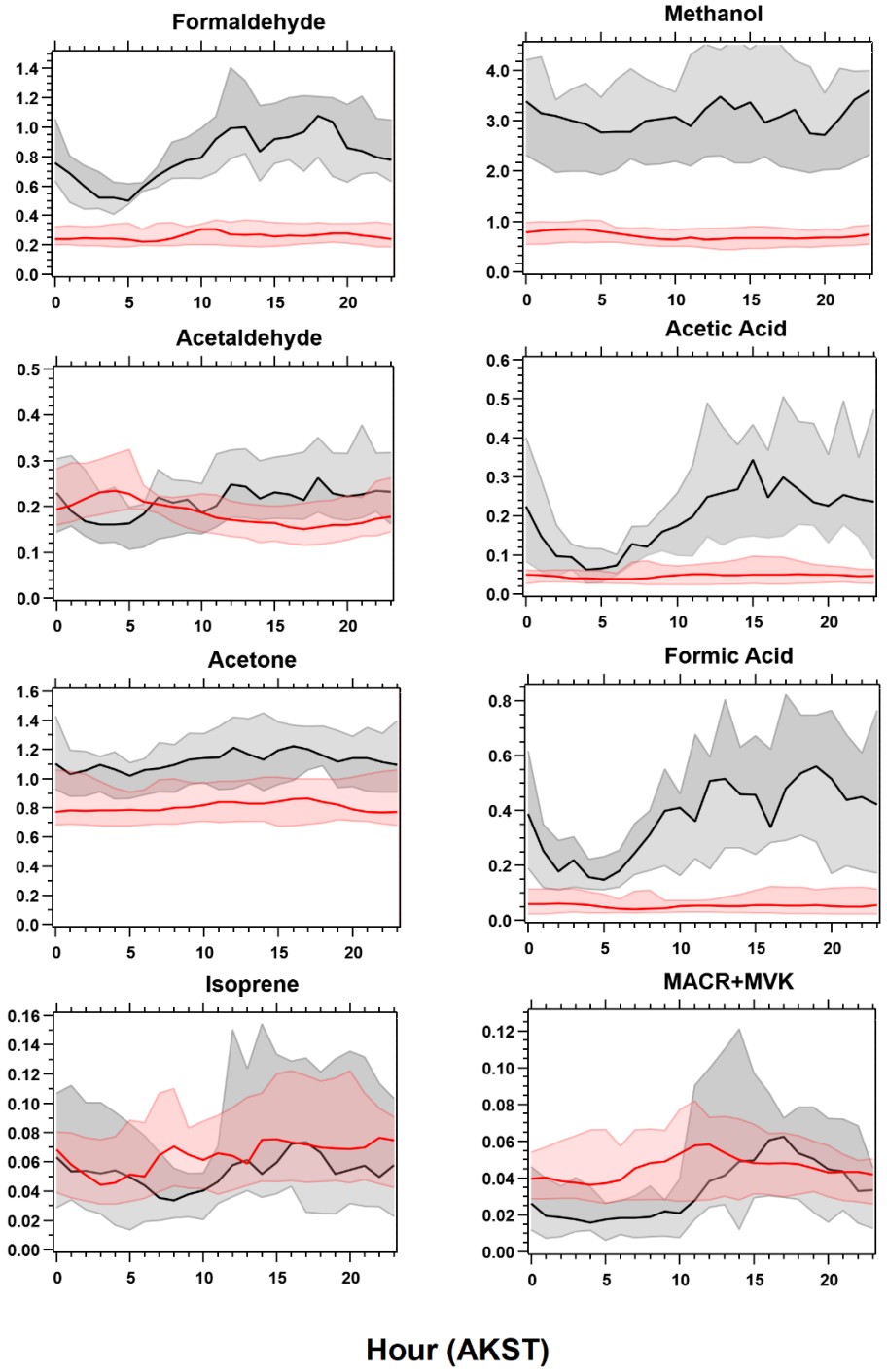

**Figure 5.** Diel cycles for select measured (black) VOCs compared to GEOS-Chem+MEGANv2.1 simulations (red). Solid lines represent median values, with shaded areas representing the 25th to 75th percentile values.

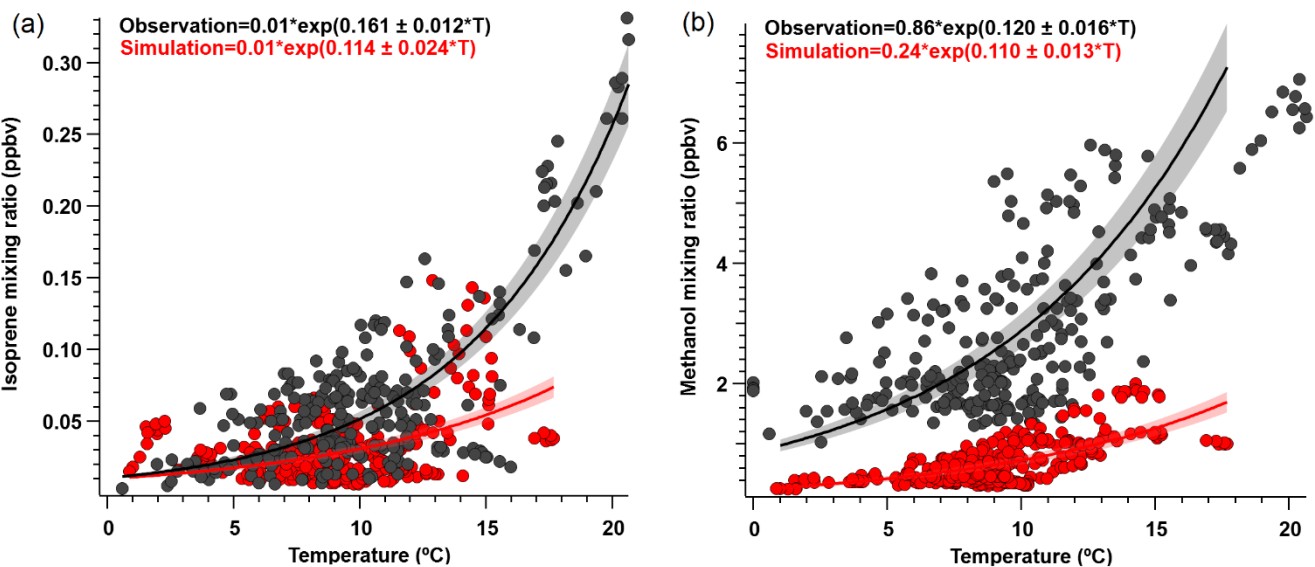


**Figure 6.** Ambient (a) isoprene and (b) methanol mixing ratio (ppbv) versus temperature (°C), for daytime values (8:00 to 20:00) where PAR > 400 µmol m$^{-2}$ s$^{-1}$. Solid lines show exponential fits (major axis regression) to observations (black) and modelled (red) outputs, following the exponential temperature activity factor in Equation 2. (Guenther et al., 2012). Shaded areas represent 95% confidence intervals. r$^2$ = >0.5 for both species.




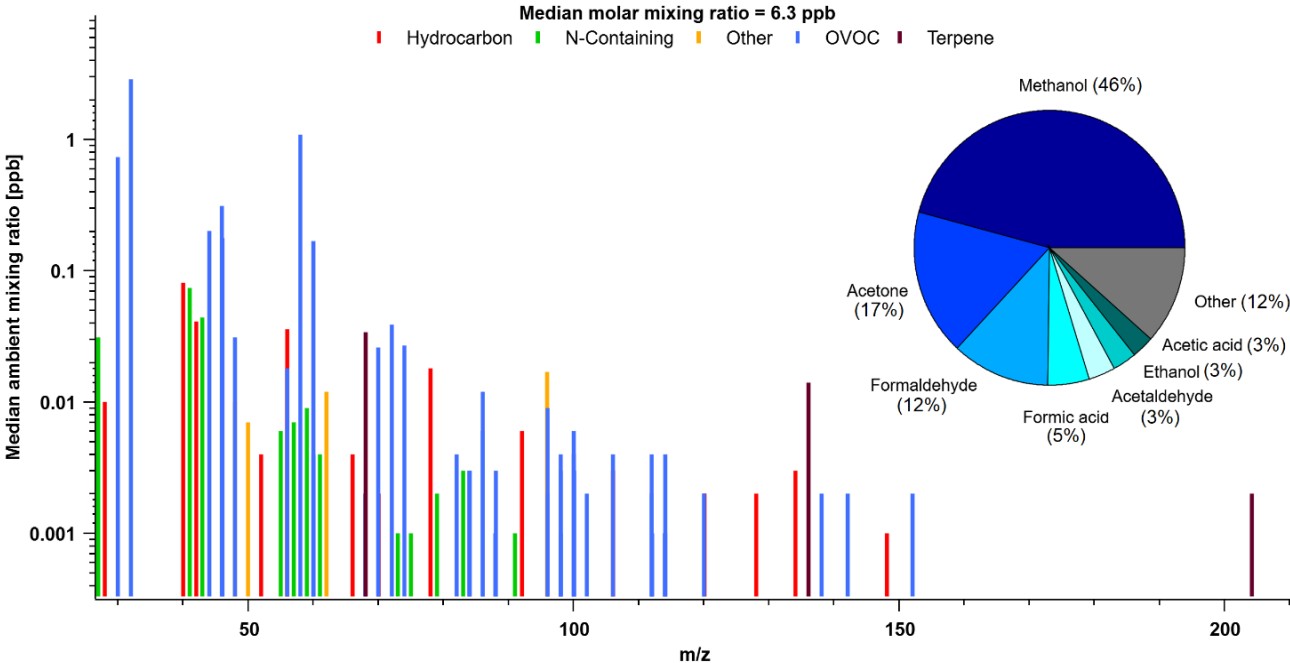

**Figure 7.** Mass spectra of PTR-ToF-MS signal ions detected and corresponding median molar mixing ratios based on hourly data collected during the campaign. Pie chart shown is the contribution from most abundant species to total (molar) VOC mass (tVOC). Ions were grouped into subcategories based on their structure and functional group. See supplemental Table 1 for subcategory assignments.

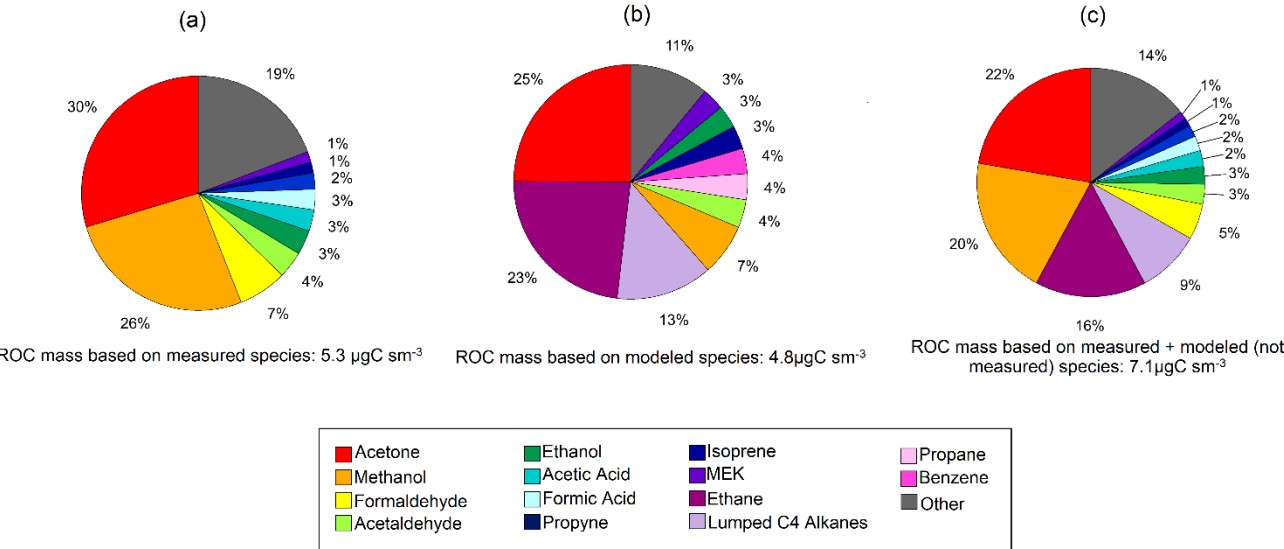

**Figure 8.** Pie charts of reactive organic carbon (ROC) for observed (a) and simulated (b) species at Toolik Field Station. Also shown is our best guess of ROC at TFS with ethane and the other higher alkanes (Lumped C4 Alkanes) estimated by the model (c). The relative contribution of individual compounds to ROC mass is calculated based on median values during the campaign.

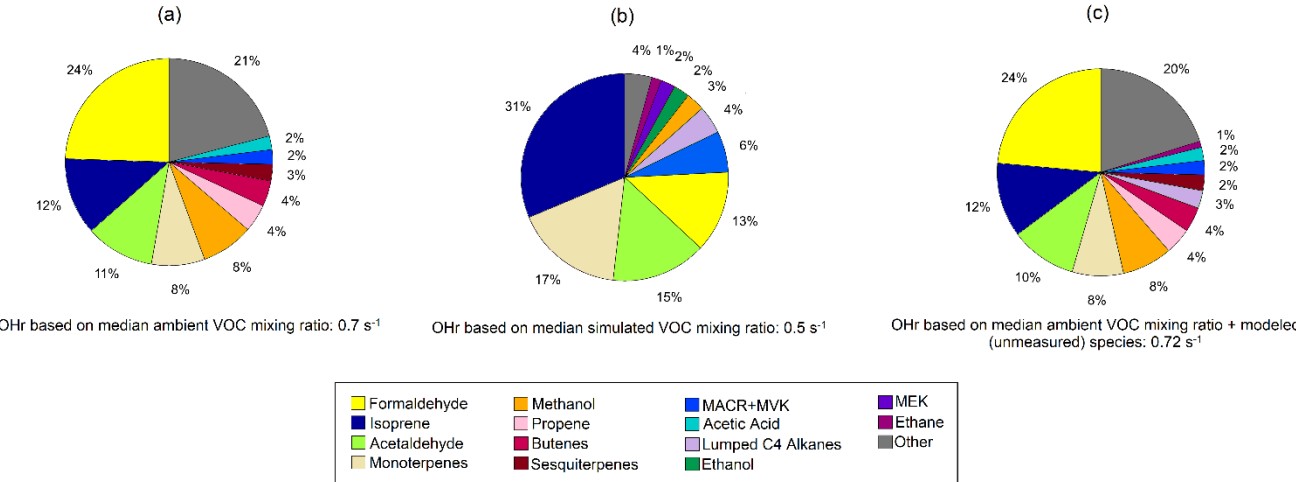

**Figure 9.** Pie charts of calculated OH reactivity (OHr) for observed (a) and simulated (b) species at Toolik Field Station. Also shown is our best guess of OHr at TFS with ethane and the other higher alkanes (Lumped C4 Alkanes) estimated by the model (c). Relative contribution of individual compounds is calculated using median campaign mixing ratios and OH rate constant for that species. Rate constants for individual VOC are compiled from previous literature, and rate constants of the dominant species or isomer at the detected PTR-ToF mass are used (Koss et al., 2018; Atkinson et al. 2004; 2006; Atkinson and Arey, 2003).