# Peer review of "Atmospheric biogenic volatile organic compounds in the Alaskan Arctic tundra: constraints from measurements at Toolik Field Station"

_Atmospheric Chemistry and Physics, 2022_

## Author Comment (AC1)

First and foremost we would like to thank the reviewers for their comments and suggestions, which have served to improve the clarity and quality of this manuscript. Our point-by-point responses to the comments below are in red, with modified manuscript text included. Apart from reviewer suggestions, we also updated the color scheme on Figure 9 to match with that of Figure 8, for ease and increased readability and comparison.

**Reviewer 1:**

The manuscript describes a very important dataset from an under-studied region for biosphere-atmosphere exchanges and surface atmosphere chemistry: the Alaskan Arctic. The presentation quality of the manuscript is exceptionally good and I only found one typo, which is quite unusual. The figures convey the results well and are clear and well captioned. All of the methodology is adequately explained or referenced appropriately. The writing throughout the manuscript is strong and impactful. The science is very important. The Arctic is ongoing rapid environmental change, driven by increasing air temperatures. This causes a cascade of effects, including shifts in ecosystem type and composition. These changes have the potential to greatly alter atmospheric chemistry in the Artic, but we currently lack adequate observations of the current state of the system. These measurements are unique and the comparison to a global chemistry model is essential. This is especially true since satellite methods of testing global chemistry models do not provide reliable data over the Arctic.

One concern with the presentation of the data is that only the early part of the growing season was captured. The measurement period ends just after the solstice, which is essentially the start of the most productive period for the tundra ecosystem. For example, isoprene concentrations are relatively low until the last week of the sample period and I would expect them to increase further in July. While this dataset is still important, the authors need to include some more cautions and caveats. As I note in the detailed comments below, this is particularly true in the conclusion section of the manuscript. While the agreement between the measurements and the model, particularly for isoprene, is encouraging, the ability of the model to predict relatively increased emissions in July is not tested.

We agree that our results are limited in this regard. Unfortunately, observational data for July was unavailable as the campaign ended before then in 2019. Additionally, data collected in August of 2018 at Toolik and shown in Fig. 3 of Angot et al., 2020 suggests that ambient isoprene concentrations only reached a maximum of ~300 pptv at the beginning of August at temperatures around ~18°C. The basal rate of isoprene emissions can be significantly affected by the air temperature from days to even weeks before the onset of emissions, and the lower maximum value in August 2018 compared to June 2019 suggests concentrations for both years were likely comparable in terms of magnitude. The average ambient air temperature at Toolik Field Station in July is typically within 1-2°C of average temperatures in June, thus we expect our results regarding the temperature dependence of ambient isoprene emissions to be comparable so long as meteorological inputs (temperature, PAR) are accurate, but higher than average maximum temperatures during this period, and large differences between surface and 2m air temperature could lead to discrepancies between measured and modeled values. Nevertheless, this is just speculation and further research is certainly warranted. We have added this caveat to our discussions and conclusions where referenced.

Also, the discrepancy with methanol is interesting, but some of the physiological aspects should be discussed. Methanol emissions are often linked to plant growth and expansion. So the sensitivity test of increasing MeOH emissions by 3x is interesting, but perhaps it's only appropriate for the early part of the growing season? This should be explored in the discussion.

It is true that methanol emissions are linked to plant growth and expansion, but methanol is also longer

lived than isoprene and not as dependent on light and temperature. Indeed, there is significant persistent underestimation in methanol emissions throughout the campaign, and the temperature response curves are statistically consistent when examining 95th percentile confidence intervals, hence why we believe this discrepancy is more related to plant functional types and base emission factors. As a result, we don't feel that this result for methanol is only limited to the early part of the growing season.

Detailed comments

Lines 140-142: Should compare your measured temperatures with long-term averages available from the Toolik weather station. Was your sampling period typical, warmer or cooler compared to the average? This is a good point and helps put our results in context. We added the following to line 142: *"A 10-year average of temperatures for this area suggest typical daily ranges of -6 ºC to 10 ºC between May and June. This range, and our campaign average, reflects the seasonal transition…"*

Line 212: Fix "accounts for considers." Done. The line now reads, *"MEGANv2.1 accounts for the major…"*

Line 228: "T is the 2m air temperature which is assumed to be equivalent to the leaf temperature." Note this is not a good assumption. This issue is discussed later in the manuscript, but some note should be made at this point about the problems with this assumption. Reviewer 2 had a similar comment. To address this, we removed the following paragraph from lines 355 to 361 (shown later in the discussion), to the methods and materials section after we address Equation 1 and 2. We also added a few references addressing the differences between air and surface temperatures.

*"Evaluation of temperature and light response within models on the effect of BVOC emissions in higher latitudes is crucial for addressing discrepancies in model simulations, as Arctic plants appear to respond to warming differently than plants from low latitudes (Rinnan et al., 2014). In addition to landscape changes in plant composition and functional type, tundra plants with relatively dark surfaces and low growth forms may also experience higher leaf temperature than air temperature measured at heights (~2 m) provided by weather stations. Studies have observed large temperature oscillations among surface vegetation (10 to 26°C), and differences of between 7-20°C when comparing air and surface temperatures (Seco et al., 2020; Lindwall et al., 2016). This could lead to larger emissions than anticipated in current models, and identified challenges in accurately estimating BVOC emissions are thus closely related to having accurate estimations of temperature and PFTs, along with representation of long-term vegetation changes (Tang et al., 2016)."*

Line 270-277: The rapid increase in isoprene concentrations (and presumably emissions) is due to both changing temperature and also phenology. Of course, temperature is ultimately driving phenology in this ecosystem. You mention phenology, which is good, but it would also be helpful to explicitly reference the observation that leaf-level isoprene emission is delayed relative to photosynthetic capacity. We added the following to the end of the sentence on line 277, *"Additionally, it is well known that the capacity for leaf-level isoprene emissions is delayed developmentally, with leaves becoming photosynthetically active weeks before isoprene emission begins. This delay is significantly affected by growth temperature, and the air temperature of previous days to weeks can affect the basal rate of isoprene emissions (Sharkey et al., 2008)."*

Line 376-377: Need to limit this conclusion and specifically note that this is only true during the

relatively cool season during which the measurements were performed. Given the non-linearities involved, 20% during this period could increase during a warmer period: for example, July.

We adjusted the paragraph per Reviewer 2's recommendation and also added the stipulation that Reviewer 1 suggested. The end of the paragraph now reads:

*"We find that despite the errors in assimilated environmental variables (T, PAR) leading to ~20% underestimation in $\gamma_T$, isoprene is only slightly (~10%) underestimated by the model (Fig. 3, 5). However, MACR+MVK is a more robust tracer to evaluate model isoprene emissions due to its longer lifetime, and decreased sensitivity in model errors due to vertical mixing, OH chemistry, or plant function type (Hu et al., 2015). Given that the errors caused by assimilated temperature and PAR inputs are minimal, we conclude that GEOS-Chem+MEGANv2.1 can reproduce regional isoprene emissions to ±20%, constrained by our observations at TFS. However, we note that our results are limited to the early growing season, and may be variable in later months (July, August), due to large discrepancies between surface and air temperatures (Seco et al., 2020; Lindwall et al., 2020). Nonetheless, better meteorological inputs can help further improve the prediction of isoprene emissions."*

Lines 460 and 474-475: Again, should note that these values for isoprene might increase later in the growing season.

*We changed Line 460 to: "Isoprene had a negligible contribution (0.5%) by comparison, but we also note that our results only capture the early part of the growing season and isoprene may therefore have a larger contribution than seen here."*

And 474 to 475: *"Interestingly, the isoprene contribution to ROC was similar (within 1%) in both observed and modeled estimates but should be further verified with measurements from later in the growing season (July, August) when there is more discrepancy between surface and air temperatures used to derive isoprene emissions."*

Line 532: Need to insert something to the effect of "during the early-season study period."

*Added "during this early-season study period" to the end of the sentence.*

**Reviewer 2:**

The manuscript by Selimovic et al. presents a comparison between VOC mixing ratios measured at the Toolik Field Station (TFS) in Alaska and mixing ratios obtained from the GEOS-Chem chemical transport model coupled with the MEGANv2.1 biogenic emission model. The total amount of reactive organic carbon (ROC) and the OH reactivity (OHr) are also compared between observations and model results.

There is a subtantial degree of uncertainty regarding the biogenic VOC emissions and the mixing ratios predicted by models, in particular in the Arctic region where there are few observations available. In this sense, the dataset and modeling exercise presented here are a first step towards filling this knowledge gap.

I must first mention that many –a lot!– of the literature references mentioned in the manuscript are missing from the reference list, which makes reviewing this manuscript very difficult. This is the first issue that the authors must address before working on a new version of the manuscript. Then, there are several other, scientific issues that need to be worked out before the manuscript can be accepted for publication, in addition to the caveats already raised by referee #1.

Regarding the references, the author used software to cite works and mistakenly uploaded an older

version contained within a previous draft of the manuscript. We apologize for the oversight. This has been updated in the manuscript, and updated references are included at the end of the response as well.

MAIN COMMENTS

Equation 1: I suspect that the excess of "expexp" is a typo. But I would like to know why the authors use a value of 200 for the coefficient named "CT2" in MEGANv2.1, while the value used in MEGANv2.1 is 230.

We thank the reviewer for pointing this out. The 'expexp' is indeed a typo. Our value for CT2 is based on the original source code for MEGANv2.1, which implemented a value of 200.

Line 232: MEGANv2.1 (Guenther et al 2012) assigns a light-dependent fraction of 80% to methanol. The authors here assign only 20%. Is this a mix-up or do the authors take these new values from some unknown source, for some unknown reasons?

It is correct that the light-dependent fraction should be 80%, while the independent fraction is 20%. This was a mix-up and we thank the reviewer for catching this. The sentence now reads:

*"On the other hand, $\gamma_T$ for methanol is computed as a weighted average of a light-dependent fraction (80%), following eqn. 1., and a light-independent fraction (20%), following eqn. 2:"...*

Line 228: Authors can say that they used the air temperature instead of the leaf temperature due to the lack of leaf temperature measurements. However, in Arctic vegetation the discrepancy between those two temperatures can be huge. See for example Lindwall et al (2016) and Seco et al (2020). Some comment about this should be included. Also in Line 243, I suggest taking with caution the stated difference of only 0.4 degrees C between the observed ambient temperature and the modeled surface temperature. Of course, the fact that the presented measurements only span the early part of the season may be the reason why the air and surface temperatures might not differ too much (the sun may not heat the surface as much as in July-August).

Reviewer 1 had a similar comment. To address this, we removed the following paragraph from lines 355 to 361 (shown later in the discussion), to the methods and materials section after we address Equation 1 and 2. We also added references addressing the differences between air and surface temperatures.

*"Evaluation of temperature and light response within models on the effect of BVOC emissions in higher latitudes is crucial for addressing discrepancies in model simulations, as Arctic plants appear to respond to warming differently than plants from low latitudes (Rinnan et al., 2014). In addition to landscape changes in plant composition and functional type, tundra plants with relatively dark surfaces and low growth forms may also experience higher leaf temperature than air temperature measured at heights (~2 m) provided by weather stations. Studies have observed large temperature oscillations among surface vegetation (10 to 26°C), and differences of between 7-20°C when comparing air and surface temperatures (Seco et al., 2020; Lindwall et al., 2016). This could lead to larger emissions than anticipated in current models, and identified challenges in accurately estimating BVOC emissions are thus closely related to having accurate estimations of temperature and PFTs, along with representation of long-term vegetation changes (Tang et al., 2016)."*

Figure 6: Isoprene temperature response is typically thought to approximately follow Equation 1 (100% light-dependent), and the authors also follow this line of though (see line 216). Why is then Equation 2 used instead in Figure 6 for the isoprene temperature response (methanol could be somewhat similar if we

follow MEGAN's 80% light-dependence for methanol)? I understand that it is easy to simply derive a beta value that can easily be compared. I would suggest to, at least, comment on this. Also, have the authors checked what is the reason that the model predicts low isoprene emissions at 17-18 degreesC while the observations are much higher in that temperature range? Is it because the real PAR conditions were much different during those timest than in the meteorological data driving the model?

We plotted Figure 6 in order to derive a beta value so that we could look at the temperature response to ambient emissions for both methanol and isoprene, and then compare that response to what the model predicted—not necessarily the values associated with MEGAN or in Guenther et al., 2012, as those are derived from flux measurements and not ambient measurements, so it's not a direct apples to apples comparison. The comparison of derived beta values using modeled and measured concentration and temperature though should be able to tell us a bit about what the model thinks isoprene's response to temperature **should** be based on assimilated temperature, compared to what it actually is when we measure. We agree with the reviewer that the isoprene temperature response should follow Equation 1, and we do plot that and make reference to it in the paragraph (E.g Fig S4). We also agree that both isoprene and methanol have some kind of light dependance associated with emissions which is why we attempted to limit that comparison to only daytime values and when PAR >400, because we knew that light wouldn't be a limiting factor in those emissions if we were to just derive a beta value for simple comparison. The reviewer is correct that the discrepancy in higher temperature emissions is due to assimilated met data. We have modified some of this paragraph and above sections for clarification.

Line 359-364, a few lines after we reference Equation 1 and Figure S4:
*"We also derive β coefficients for isoprene and methanol to determine the temperature response of emissions, with higher β indicating a steeper temperature response curve and vice versa. Isoprene and methanol both exhibit light dependence, thus we controlled for this by ionly looking at $\gamma_T$ during daytime hours (08:00 to 20:00) and when PAR was >400 $\mu mol\ m^{-2}\ s^{-1}$."*

L 366-367: *"However, this may also be partially due to differences in observed versus assimilated meteorology during some of the warmest days. Additionally,…"*

Line 373: I cannot see how the 20% discrepancy in PAR (the authors do not say the sign of the discrepancy but in Fig. 1c the modeled PAR is clearly underestimated by the model, at least on average) can make up for an underestimation of the temperature activity factor, because the light activity factor will be understimated as well. The rest of the paragraph needs some redoing since I found it confusing. Why is MCAR+MVK a more robust tracer to avaluate model isoprene emission?

MACR+MVK is a more robust tracer to evaluate model isoprene emissions because both are oxidation products of isoprene, but are longer lived than isoprene, and therefore less sensitive to model errors in vertical mixing, OH chemistry, plant function types, etc. We've added this clarification. We also rewrote the paragraph beginning in (original) line numbers 372. The rest of the paragraph now reads:

*"We find that despite the errors in assimilated environmental variables (T, PAR) leading to ~20% underestimation in $\gamma_T$, isoprene is only slightly (~10%) underestimated by the model (Fig. 3, 5). However, MACR+MVK is a more robust tracer to evaluate model isoprene emissions due to its longer lifetime, and decreased sensitivity in model errors due to vertical mixing, OH chemistry, or plant function type (Hu et al., 2015). Given that the errors caused by assimilated temperature and PAR inputs are minimal, we conclude that GEOS-Chem+MEGANv2.1 can reproduce regional isoprene emissions to ±20%, constrained by our observations at TFS. However, we note that our results are limited to the early growing season, and may be variable in later months (July, August), due to large discrepancies between surface and air temperatures (Seco et al., 2020; Lindwall et al., 2020). Nonetheless, better*

*meteorological inputs can help further improve the prediction of isoprene emissions."*

Line 419: do the authors have any information about methane mixing ratios at TFS to be able to confidently exclude that methane oxidation is an important source of formaldehyde?
Unfortunately we don't have accurate estimates, but methane is typically long lived and shouldn't vary too much. If methane oxidation was an importance source of formaldehyde, one would expect baseline errors in formaldehyde emissions to be incorrect since it's persistent in the background. In our case, adding baseline concentrations doesn't necessarily improve the agreement between observed and modeled ambient concentrations, thus we don't believe it's a primary source of formaldehyde.

MINOR POINTS

Lines 26-27: I found the sentence about isoprene unclear. Please rephrase.
We changed the line to *"Isoprene was the most abundant terpene identified."*

Line 212: "accounts for considers" should be either "accounts for" or "considers"
Done. The line now reads, *"MEGANv2.1 accounts for the major…"*

Line 253: I suggest replacing "masses" with "compounds"
Done.

Line 349: "only" is duplicated
Removed duplicate "only."

Line 365: "9:00" should be "21:00"
Changed and fixed.

Line 377: is the sentence ending in this line referring to Fig. 5? If so, please state it.
This was in reference to both Figure 3 and 5. The appropriate reference to these figures has been added.

Line 380: The "comparisons of measured versus simulated" are shown in Fig. 3? Or Fig. 5? Or both? Please clarify this in this sentence to guide the reader.
Changed to *"comparisons of measured versus simulated OVOC abundance shown in Figs. 3 and 5."*

Line 474: should the last word be "modeled" instead of "measured"?
Correct. Changed to *"observed and modeled estimates"*

Line 501: I suggest using past tense
Done. Changed *"comes from formaldehyde…"* to *"came from formaldehyde..."*

Figure S5: This is not mentioned in the text, but I am curious about the last (right-most) measured datapoints of the graph. Shouldn't the air be well mixed at that point of the day (as the model suggests and the authors point out for the left side of the graph)? Is the high red point (0m) due to emission by vegetation at the surface?
This is a correct assessment. The high red point is indeed due to emission by vegetation at the surface.

References

Guenther, A., Karl, T., Harley, P., Wiedinmyer, C., Palmer, P.I., and Geron, C.: Estimates of global terrestrial isoprene emissions using MEGAN (
[revised manuscript text omitted]